# PRETRAINING TEXT ENCODERS WITH ADVERSARIAL MIXTURE OF TRAINING SIGNAL GENERATORS

**Yu Meng**[1]* **Chenyan Xiong**[2] **Payal Bajaj**[2] **Saurabh Tiwary**[2] **Paul Bennett**[2]
**Jiawei Han**[1] **Xia Song**[2]
[1]University of Illinois at Urbana-Champaign [2]Microsoft
[1]{yumeng5,hanj}@illinois.edu [2]{chenyan.xiong,payal.bajaj,
satiwary,paul.n.bennett,xiaso}@microsoft.com

## ABSTRACT

We present a new framework AMOS that pretrains text encoders with an Adversarial learning curriculum via a Mixture Of Signals from multiple auxiliary generators. Following ELECTRA-style pretraining, the main encoder is trained as a discriminator to detect replaced tokens generated by auxiliary masked language models (MLMs). Different from ELECTRA which trains one MLM as the generator, we jointly train multiple MLMs of different sizes to provide training signals at various levels of difficulty. To push the discriminator to learn better with challenging replaced tokens, we learn mixture weights over the auxiliary MLMs' outputs to maximize the discriminator loss by backpropagating the gradient from the discriminator via Gumbel-Softmax. For better pretraining efficiency, we propose a way to assemble multiple MLMs into one unified auxiliary model. AMOS outperforms ELECTRA and recent state-of-the-art pretrained models by about 1 point on the GLUE benchmark for BERT base-sized models.

## 1 INTRODUCTION

Instead of pretraining encoders on texts with random noise (*e.g.*, randomly masked tokens in BERT (Devlin et al., 2019)), ELECTRA-style frameworks employ an auxiliary model to corrupt the original text sequence using its language modeling outputs, and pretrain the main Transformer to detect (Clark et al., 2020) or correct (Meng et al., 2021) the replaced tokens. These new self-supervised frameworks significantly improve the efficiency and effectiveness of pretraining and lead to strong results in various downstream tasks with fine-tuning (Clark et al., 2020), prompt-based learning (Meng et al., 2021), and zero-shot cross-lingual representations (Chi et al., 2021).

Recent studies revealed that the key to the ELECTRA's success is its new learning dynamics (Xu et al., 2020; Meng et al., 2021). By pretraining the auxiliary model jointly with the main Transformer, an implicit learning curriculum is formed: The noise produced by the auxiliary generator becomes more and more plausible during pretraining, posing greater challenges for the discriminator, which has to overcome the difficulty by reasoning more deeply using the contexts. This leads to significantly improved sample efficiency and effectiveness of ELECTRA-style pretrained models (Clark et al., 2020; Chi et al., 2021; Meng et al., 2021).

On the other hand, such a training dynamic also introduced new challenges in search of the optimal pretraining setting. First, the configurations of the auxiliary generator—its depth, width, and masking fraction—require costly trial-and-error pretraining runs. At the same time, they also significantly impact the discriminator's downstream task performance: A weak auxiliary model does not generate hard enough pretraining signal to push the discriminator, but a too strong one can confuse the discriminator and worsen its downstream task performance (Clark et al., 2020; Meng et al., 2021). Second, the side-by-side training of the two models forms a pseudo "GAN-style" (Goodfellow et al., 2014) curriculum which causes difficulty to improve or scale: Previous attempts to make the generator and discriminator learning more interactive (*e.g.*, training the generator to maximize the discriminator loss as in actual GAN frameworks) resulted in downgraded performance (Clark et al., 2020).

---

*Part of this work was done while Yu was interning at Microsoft.

In this paper, we present a new approach that learns to automatically select pretraining signals and constructs the learning curriculum of ELECTRA-style frameworks. Our approach, AMOS, "*A*dversarial *M*ixture *O*f *S*ignals" for text encoder pretraining, samples a diverse set of pretraining signals from different layers of an auxiliary generator. The sampling is conducted with Gumbel-Softmax gradient estimation of the (reversed) gradient backpropagated from the main discriminator, which allows adversarial learning of mixture weights over the auxiliary generator's signals. In this way, AMOS constructs a more diverse set of pretraining signals for the main discriminator training and enables an automatic selection of the signals to form a more effective learning curriculum. In addition, by sampling from different layers of one generator model, AMOS ensures minimum computation overhead.

Our experiments on the GLUE and SQuAD benchmark demonstrate the effectiveness of AMOS. In the standard $BERT_{Base}$ and $RoBERTa_{Base++}$ pretraining settings (Devlin et al., 2019), AMOS shows robust advantage across all included language representation tasks. The improvements are significant as it advances the state-of-the-arts by about 1 absolute point on average GLUE score and has competitive SQuAD accuracy. Our thorough ablation studies confirm that such effectiveness stems from the diverse training signals from the mixture of generators and the adversarial learning approach to combine them.[1]

In the rest of this paper, we first recap related work (Section 2) and then propose our methods (Section 3). After that we describe our experimental settings (Section 4) and evaluation results (Section 5). The last section concludes and discusses potential future research directions.

## 2 RELATED WORK

Besides the standard auto-regressive language modeling (Radford et al., 2019) and masked language modeling (MLM) (Devlin et al., 2019), many have explored different designs of the pretraining task, for example, prefix language modeling (Raffel et al., 2019), permutational language modeling (Yang et al., 2019), and unified language modeling (Dong et al., 2019). The advantage of *manually constructed* pretraining tasks is more often observed in an application-specific manner, where prior knowledge about the target scenario is introduced (Guu et al., 2020; Roberts et al., 2020; Jia et al., 2021; Lu et al., 2021). To pretrain general purpose language representations, the standard MLM task often offers the best combination of simplicity, robustness, and generalization ability among those manually constructed pretraining signals (Raffel et al., 2019; Lewis et al., 2019).

Clark et al. (2020) developed a novel framework ELECTRA to pretrain Transformers with *model-generated* signals. They employed an auxiliary MLM model, the "generator", to replace the masked positions in the input sequence with tokens sampled from its language model head. The main Transformer, the "discriminator", is pretrained to detect the replaced tokens. In this way, the main model is trained to the denoise the more challenging noises from the auxiliary language model. As a result, ELECTRA requires much fewer pretraining data points to reach the performance of MLM pretrained models, and offers better downstream performance when converged.

The unique effectiveness of ELECTRA intrigued many studies to understand its real source of advantage. Clark et al. (2020) originally noted the sample efficiency comes from ELECTRA's replaced token detection task being able to leverage all token positions for training. Later, Xu et al. (2020) pointed to the more influential change of the pretraining task complexity for the main Transformer. Recently, Meng et al. (2021) conducted a thorough ablation study on many variants of ELECTRA-style models and revealed the benefits of pretraining with more challenge signals generated by the auxiliary model, as well as the implicit learning curriculum by pretraining two models side-by-side.

One popular line of research to improve ELECTRA-style models is to upgrade the replaced token detection task to more semantically informative ones. Xu et al. (2020) proposed the multi-word choice task which pretrains the main model to pick the original token from a small candidate set. COCO-LM developed a corrective language modeling task which pretrains the main model to recover the original tokens (Meng et al., 2021).

---

[1]Code and pretrained models can be found at `https://github.com/microsoft/AMOS`.

Another line of research is to improve the training signal construction on the auxiliary side. In ELECTRA, Clark et al. (2020) experimented with adversarially training the auxiliary model using feedback from the discriminator model via reinforcement learning, but observed worse downstream performance. Hao et al. (2021) kept the auxiliary model unchanged and augmented the replaced token sampling with predicted signal difficulty and smoothed probability via Focal Loss (Lin et al., 2017). In this work, we continue this line of research and focus on how to more automatically and effectively construct the learning signals for ELECTRA-style models.

## 3 METHOD

In this section, we first recap the preliminary of ELECTRA-style pretraining, the explorations of its learning curriculum, and then present our AMOS framework.

### 3.1 PRELIMINARY ON ELECTRA-STYLE PRETRAINING

In ELECTRA-style pretraining (Clark et al., 2020), two Transformer models are trained side-by-side: One is trained via standard masked language modeling (MLM) and is used to generate replaced tokens to corrupt the original sequences; the other is trained to denoise the corruptions from the first model (*e.g.*, via replaced token detection (RTD) (Clark et al., 2020), multi-word choice (Xu et al., 2020), or corrective language modeling (Meng et al., 2021)). The second model is the main model to use in downstream tasks (the "discriminator"). The first one is the auxiliary, often referred to as the training data "generator". In this work we built upon the original ELECTRA setup where the main model is trained with RTD.

**Generator Training.** Given an original sequence $X^{\mathrm{orig}} = [x_1^{\mathrm{orig}}, \ldots, x_i^{\mathrm{orig}}, \ldots, x_n^{\mathrm{orig}}]$, 15% of its tokens are randomly replaced by [MASK] symbols, and the resulting masked sequence $X^{\mathrm{mask}} = [x_1^{\mathrm{orig}}, \ldots, \mathtt{[MASK]}_i, \ldots, x_n^{\mathrm{orig}}]$ is fed to the generator which is trained via the following loss to predict the original tokens from the vocabulary $V$ at the set of masked positions $\mathcal{M}$:

$$p_{\mathrm{MLM}}(x_t|\boldsymbol{h}_i) = \frac{\exp(\boldsymbol{x}_t^\top \boldsymbol{h}_i)}{\sum_{t'=1}^{|V|} \exp(\boldsymbol{x}_{t'}^\top \boldsymbol{h}_i)}; \quad \mathcal{L}_{\mathrm{GEN}} = \mathbb{E}\left(-\sum_{i \in \mathcal{M}} \log p_{\mathrm{MLM}}\left(x_i^{\mathrm{orig}}|\boldsymbol{h}_i\right)\right),$$

where $\{\boldsymbol{h}_i\}_{i=1}^n$ are the contextualized representations generated by the Transformer (after the projection head) and $\{\boldsymbol{x}_t\}_{t=1}^{|V|}$ are the token embeddings.

**Discriminator Training.** A replaced sequence $X^{\mathrm{rep}}$ is constructed by sampling from the generator's MLM probability distribution:

$$x_i^{\mathrm{rep}} \sim p_{\mathrm{MLM}}\left(x|\boldsymbol{h}_i\right), \text{ if } i \in \mathcal{M} ; \quad x_i^{\mathrm{rep}} = x_i^{\mathrm{orig}}, \text{ else.}$$

The discriminator takes $X^{\mathrm{rep}}$ as input and is trained to distinguish replaced tokens produced by the generator against the original tokens via the binary classification loss:

$$\mathcal{L}_{\mathrm{DISC}} = \mathbb{E}\left(-\sum_{x_i^{\mathrm{rep}}=x_i^{\mathrm{orig}}} \log p_{\mathrm{RTD}}\left(x_i^{\mathrm{rep}} = x_i^{\mathrm{orig}}\big|\boldsymbol{h}_i\right) - \sum_{x_i^{\mathrm{rep}}\neq x_i^{\mathrm{orig}}} \log\left(1 - p_{\mathrm{RTD}}\left(x_i^{\mathrm{rep}} = x_i^{\mathrm{orig}}\big|\boldsymbol{h}_i\right)\right)\right),$$

$$\tag{1}$$

where $p_{\mathrm{RTD}}(x_i^{\mathrm{rep}} = x_i^{\mathrm{orig}}|\boldsymbol{h}_i) = \exp(\boldsymbol{w}^\top \boldsymbol{h}_i)/(1 + \exp(\boldsymbol{w}^\top \boldsymbol{h}_i))$ and $\boldsymbol{w}$ is a learnable weight vector.

By training the generator side-by-side with the discriminator, this ELECTRA-Style framework constructs a natural learning curriculum for the main model. At the beginning, the generator is weak and the replaced tokens are nearly random, making the pretraining task of the main model easy for a head start. As the generator learns better, the corruptions from it become more challenging, which increase the difficulty of the learning task for the main model. After a series of studies, recent research confirmed that this curriculum learning is the key to ELECTRA-style framework's effectiveness (Clark et al., 2020; Xu et al., 2020; Meng et al., 2021). Pretraining with randomly replaced tokens or starting with a converged generator's harder signals both significantly worsen the discriminator effectiveness (Meng et al., 2021).

## 3.2 IN SEARCH OF THE OPTIMAL CURRICULUM IN ELECTRA

In general, configuring a learning curriculum for neural model training is tricky (Bengio et al., 2009; Hacohen & Weinshall, 2019). Many moving pieces are involved—how to measure the difficulty of training signals; when and how to change the training signal difficulty—all require manual trial-and-error and careful designs (Graves et al., 2017). The same challenge persists in ELECTRA-style pretraining models. More specifically, it is unclear how to choose the size of the generator network and how to control the training dynamics of the generator and discriminator.

**Generator Size.** There are many factors on the auxiliary side that affect the difficulty of the generated signals: Network depth, width, MLM mask fraction, and whether to use dropout when sampling the replaced tokens. All of these have notable impact on the performance of the pretrained discriminator model. The general consensus is that the generator should be smaller than the discriminator but all the rest vary setting by setting.

The original ELECTRA framework kept the generator's Transformer layer depth the same with the discriminator model, enabled dropout in both its

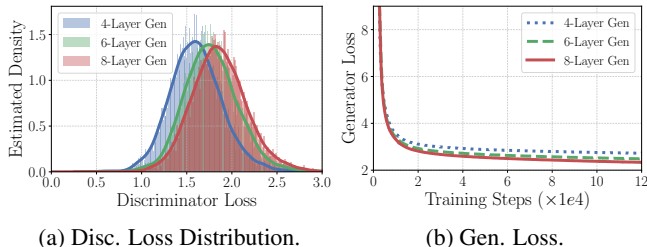

(a) Disc. Loss Distribution.    (b) Gen. Loss.

Figure 1: (a) After pretraining: Distribution of discriminator loss on replaced tokens by $4/6/8$-layer generators. Histograms/Curves are distribution bins/kernel density estimates. (b) During pretraining: $4/6/8$-layer generator loss.

training and sampling, and mainly explored using a smaller width (*e.g.*, with $1/3$ hidden dimensions of the main model). They also empirically found that large-sized discriminators prefer different mask fractions on the generator side.

Later, COCO-LM (Meng et al., 2021) empirically found that it is better to use a shallower generator but of the same width as the main model, while also disabling dropout when sampling replaced tokens. The depth of the generator is still chosen empirically and the optimal settings vary with the configurations of the discriminator model.

To illustrate the difficulty of training signals by generators with different depths, we train BERT$_{\text{Base}}$-sized discriminators with $4/6/8$-layer generators, and show their RTD loss on replaced tokens by these different-sized generators in Figure 1a. The average discriminator loss grows with generator depth, indicating the increased signal difficulty, and the three distributions overlap with each other. The current practice is to enumerate different generators and choose the one leading to the best downstream task performance of the discriminator, which is tedious and unsustainable.

**Generator Training Dynamics.** As discussed earlier, ELECTRA constructs an implicit learning curriculum with the training process of the generator. To illustrate this implicit curriculum, in Figure 1b we plot the MLM loss of the generators during pretraining. The convergence of the MLM models follows a logarithmic schedule: The loss drops sharply in the first $10\%$ of pretraining and decreases slowly afterwards. It is unclear whether such a training process of the generator automatically leads to the optimal curriculum for discriminator training. Unfortunately, it is challenging to manually explore different generator training dynamics in ELECTRA-style frameworks as it is trained side-by-side with the discriminator.

One idea to improve ELECTRA-style frameworks is to connect the training of the generator with feedbacks from the discriminator (*e.g.*, as in GAN-style models (Goodfellow et al., 2014)) so that the generator can be adapted based on the latest discriminator's state throughout pretraining. Clark et al. (2020) experimented with training the generators to produce more difficult signals for the discriminator, using policy gradients from the latter. However, this adversarial setup makes the training unstable and yields worse results, similar to the instability of GAN-style training in other language modeling tasks (Caccia et al., 2020). Hao et al. (2021) introduced a training signal difficulty estimator to adjust the sampling of replaced tokens in MLM for more challenging training signals. It achieved better results than the original ELECTRA but still kept the generator training intact.

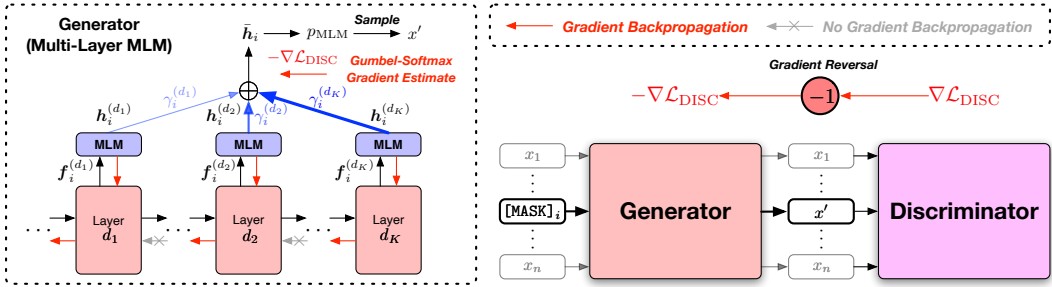

Figure 2: Overview of AMOS. The generator has multiple layers trained with MLM to provide training signals of various levels of difficulty. The mixture weights over MLM outputs are learned to maximize the discriminator loss, by backpropagating the estimated reversed gradient from the discriminator via Gumbel-Softmax. The discriminator is trained by the RTD task.

## 3.3 ADVERSARIAL LEARNING WITH MIXTURE OF GENERATORS

Intuitively, different-sized generators provide training signals at different levels of difficulty and also may advocate the discriminator to capture different linguistic information. For example, as shown in various probing studies (Tenney et al., 2019a;b), a very shallow MLM model may still make trivial syntactic mistakes, thus the discriminator can focus on learning simple language syntax to identify these errors, while the errors made by a deep MLM model may require the discriminator to capture more sophisticated language semantics to detect.

In this work, instead of empirically searching for the best generator setup, we propose the AMOS framework that (1) employs multiple MLM generators to construct a diverse set of pretraining signals, and (2) adversarially learns the mixture of these generator outputs from discriminator feedback to construct more challenging signals. These two designs enable AMOS to automatically compose an effective curriculum. Figure 2 shows an overview of our AMOS framework.

**Multi-Layer MLM Generator.** A straightforward way to obtain a diverse set of signals for replaced token generation is to utilize multiple generator networks of different depths. However, jointly training $K$ different MLM generators is expensive. Therefore, we propose to train a single generator with multiple MLM heads at different layers to mimic the effect of using multiple MLM generators. Specifically, suppose we aim to use $K$ generators of depths $\mathcal{D} = \{d_1, d_2, \ldots, d_K\}$ ($d_1 < d_2 < \cdots < d_K$), then we will instead train one generator of depth $d_K$, and insert an MLM head to each layer $d \in \mathcal{D}$. The $K$ MLM heads will be jointly trained and the MLM head parameters are shared across the $K$ heads:

$$p_{\text{MLM}}\left(x_t\big|\boldsymbol{h}_i^{(d)}\right) = \frac{\exp\left(\boldsymbol{x}_t^\top \boldsymbol{h}_i^{(d)}\right)}{\sum_{t'=1}^{|V|} \exp\left(\boldsymbol{x}_{t'}^\top \boldsymbol{h}_i^{(d)}\right)}; \quad \mathcal{L}_{\text{GEN}} = \mathbb{E}\left(-\sum_{i \in \mathcal{M}} \sum_{d \in \mathcal{D}} \log p_{\text{MLM}}\left(x_i^{\text{orig}}\big|\boldsymbol{h}_i^{(d)}\right)\right), \quad (2)$$

where $\{\boldsymbol{h}_i^{(d)}\}_{i=1}^n$ are the contextualized representations generated by the Transformer at the $d$th layer (after the projection head). During training, gradient backpropagation from upper layers is disabled at each layer $d \in \mathcal{D}$ so that the generator is partitioned into $K$ blocks where each block is trained via its own MLM objective without being disturbed by other blocks. This allows each block to act as individual MLM generators, while also leveraging the representations generated by previous blocks.

**Learning Adversarial Mixture Weights.** It would be desirable to progressively adapt the output signals of the multi-layer MLM generator based on the discriminator's state throughout pretraining. The generated replaced tokens are thus more informative for discriminator training (*i.e.*, avoid generating easy signals which can already be well distinguished by the discriminator). Given the instability of GAN-style training to language modeling (Caccia et al., 2020; Clark et al., 2020), we maintain the MLM training objective of the generator, and adjust the outputs of the generator by learning mixture weights over the multiple MLM heads' outputs.

Specifically, for each masked position $i \in \mathcal{M}$, we learn mixture weights $\boldsymbol{\gamma}_i$ over the embeddings $\boldsymbol{h}_i^{(d)}$, which is generated by each MLM head, to obtain the combined MLM embedding $\bar{\boldsymbol{h}}_i$. Then the

final token sampling probability distribution $p_{\text{MLM}}$ is computed:

$$\gamma_i^{(d)} = \frac{\exp\left(\boldsymbol{v}^\top \boldsymbol{f}_i^{(d)}\right)}{\sum_{d' \in \mathcal{D}} \exp\left(\boldsymbol{v}^\top \boldsymbol{f}_i^{(d')}\right)}; \ \bar{\boldsymbol{h}}_i = \sum_{d \in \mathcal{D}} \gamma_i^{(d)} \boldsymbol{h}_i^{(d)}; \ p_{\text{MLM}}\left(x_t | \bar{\boldsymbol{h}}_i\right) = \frac{\exp\left(\boldsymbol{x}_t^\top \bar{\boldsymbol{h}}_i\right)}{\sum_{t'=1}^{|V|} \exp\left(\boldsymbol{x}_{t'}^\top \bar{\boldsymbol{h}}_i\right)}, \quad (3)$$

where $\boldsymbol{v}$ is a learnable weight vector; $\{\boldsymbol{f}_i^{(d)}\}_{i=1}^n$ are the contextualized representations generated by the Transformer at the $d$th layer (before the projection head in MLM).

However, directly sampling from the above MLM distribution fails to enable the mixture weight learning to be guided by the discriminator: The sampling operation is non-differentiable and prevents gradient backpropagation from the discriminator to the generator. We leverage Gumbel-Softmax (Jang et al., 2017) for a continuous approximation to the sampling operation for gradient estimation:

$$\hat{p}_{\text{MLM}}\left(x_t | \bar{\boldsymbol{h}}_i\right) = \frac{\exp\left((\log \pi_t + g_t)/\tau\right)}{\sum_{t'=1}^{|V|} \exp\left((\log \pi_{t'} + g_{t'})/\tau\right)}; \quad \forall t, \ \pi_t = p_{\text{MLM}}\left(x_t | \bar{\boldsymbol{h}}_i\right), \ g_t \sim \text{Gumbel}(0, 1),$$

where $\tau$ is a temperature hyperparameter (a smaller $\tau$ results in a more accurate approximation).

Once we have the gradient estimation backpropagated from the discriminator side, we will update the mixture weights $\boldsymbol{\gamma}$ to *maximize* the discriminator loss. This forms an *adversarial* learning curriculum as the multiple MLMs' outputs are always combined to construct challenging replaced tokens given the latest discriminator state. Such an adversarial learning setup can be easily implemented via a gradient reversal layer (Ganin et al., 2016) that multiplies the gradient backpropagated from the discriminator side by $-1$ when updating the mixture weights.

**Overall Training.** AMOS jointly trains the multi-layer MLM generator $\boldsymbol{\theta}_{\text{GEN}}$, the mixture weight parameter $\boldsymbol{v}$, and the discriminator $\boldsymbol{\theta}_{\text{DISC}}$ via the following losses:

$$\boldsymbol{\theta}_{\text{GEN}}^*, \boldsymbol{v}^* \leftarrow \arg\min_{\boldsymbol{\theta}_{\text{GEN}}, \boldsymbol{v}} \left(\mathcal{L}_{\text{GEN}} - \lambda \mathcal{L}_{\text{DISC}}\right), \quad \boldsymbol{\theta}_{\text{DISC}}^* \leftarrow \arg\min_{\boldsymbol{\theta}_{\text{DISC}}} \mathcal{L}_{\text{DISC}}, \quad (4)$$

where $\lambda$ is a hyperparameter balancing the weight of the two losses. Similar to ELECTRA, the generator minimizes $\mathcal{L}_{\text{GEN}}$ in Eqn. (2) and the discriminator minimizes $\mathcal{L}_{\text{DISC}}$ in Eqn. (1). The notable difference from ELECTRA is that AMOS additionally trains the generator's mixture weights $\boldsymbol{\gamma}$ (thus $\boldsymbol{v}$ and $\boldsymbol{\theta}_{\text{GEN}}$) to maximize the discriminator loss. Note that the gradient backpropagated from the discriminator through the Gumbel-Softmax estimation is used to update the mixture weights, but *not* the MLM embeddings (*i.e.*, in Eqn. (3), $\gamma_i^{(d)}$ is updated by the gradient from the discriminator side, while $\boldsymbol{h}_i^{(d)}$ is not). This guarantees that the generator's language modeling ability is still acquired through the MLM task without being disturbed by discriminator signals.

## 4 EXPERIMENTAL SETUP

**Pretraining Settings.** We experiment with two standard pretraining settings, *base* and *base++*: *Base* is the BERT$_{\text{Base}}$ training configuration (Devlin et al., 2019): Pretraining on Wikipedia and BookCorpus (Zhu et al., 2015) (16 GB of texts) for 256 million samples on 512 token sequences (125K batches with 2048 batch size). We use the same corpus and 32,768 uncased BPE vocabulary (Sennrich et al., 2015) as with TUPE (Ke et al., 2020) and COCO-LM (Meng et al., 2021).

*Base++* trains the base size model with larger corpora and/or more training steps. We add in OpenWebText (Gokaslan & Cohen, 2019), CC-News (Liu et al., 2019) and STORIES (Trinh & Le, 2018), to a total of 160 GB texts, and train for 4 billion (with 2048 batch size) samples, following recent research Bao et al. (2020); Liu et al. (2019); Yang et al. (2019). We follow the prepossessing of UniLMV2 (Bao et al., 2020) and use 64,000 cased BPE vocabulary.

**Model Architecture.** Our *base/base++* discriminator model uses the BERT$_{\text{Base}}$ architecture (Devlin et al., 2019): 12 layer Transformer, 768 hidden size, plus T5 relative position encoding (Raffel et al., 2019). Our generator is an 8-layer Transformer with the same hidden size, and the MLM heads are inserted at the 4th, 6th and 8th layers (to mimic having three generators). We disable dropout in the generator following Meng et al. (2021).

**Downstream Tasks.** We use the tasks included in GLUE (Wang et al., 2018) and SQuAD 2.0 reading comprehension (Rajpurkar et al., 2016). All models are evaluated with the same standard fine-tuning

Table 1: Results on GLUE and SQuAD 2.0 development set (GLUE test set results can be found in Appendix D). All results are single-task, single-model fine-tuning. We use Spearman correlation for STS, Matthews correlation for CoLA, and accuracy for the rest on GLUE. AVG is the average of the eight tasks on GLUE. All baseline results are reported by previous research. Results not available in public reports are marked as "–".

| Model | Params | GLUE DEV Single Task | | | | | | | | | SQuAD 2.0 | |
| | | MNLI | QQP | QNLI | SST-2 | CoLA | RTE | MRPC | STS-B | AVG | EM | F1 |
|---|---|---|---|---|---|---|---|---|---|---|---|---|
| **Base Setting:** BERT Base Size, Wikipedia + Book Corpus (16GB) | | | | | | | | | | | | |
| BERT (Devlin et al., 2019) | 110M | 84.5/- | 91.3 | 91.7 | 93.2 | 58.9 | 68.6 | 87.3 | 89.5 | 83.1 | 73.7 | 76.3 |
| RoBERTa (Liu et al., 2019) | 110M | 85.8/85.5 | 91.3 | 92.0 | 93.7 | 60.1 | 68.2 | 87.3 | 88.5 | 83.3 | 77.7 | 80.5 |
| XLNet (Yang et al., 2019) | 110M | 85.8/85.4 | – | – | 92.7 | – | – | – | – | – | 78.5 | 81.3 |
| DeBERTa (He et al., 2021) | 134M | 86.3/86.2 | – | – | – | – | – | – | – | – | 79.3 | 82.5 |
| TUPE (Ke et al., 2020) | 110M | 86.2/86.2 | 91.3 | 92.2 | 93.3 | 63.6 | 73.6 | 89.9 | 89.2 | 84.9 | – | – |
| ELECTRA (Clark et al., 2020) | 110M | 86.9/86.7 | 91.9 | 92.6 | 93.6 | 66.2 | 75.1 | 88.2 | 89.7 | 85.5 | 79.7 | 82.6 |
| +HP$_{Loss}$+Focal (Hao et al., 2021) | 110M | 87.0/86.9 | 92.7 | 91.7 | 92.6 | 66.7 | 90.7 | 81.3 | 91.0 | 86.7 | 83.0 | 85.6 |
| MC-BERT (Xu et al., 2020) | 110M | 85.7/85.2 | 89.7 | 91.3 | 92.3 | 62.1 | 75.0 | 86.0 | 88.0 | 83.7 | – | – |
| COCO-LM (Meng et al., 2021) | 110M | 88.5/88.3 | 92.0 | 93.1 | 93.2 | 63.9 | 84.8 | **91.4** | 90.3 | 87.2 | 82.4 | 85.2 |
| AMOS | 110M | **88.9/88.7** | **92.3** | **93.6** | **94.2** | **70.7** | **86.6** | 90.9 | **91.6** | **88.6** | **84.2** | **87.2** |
| **Base++ Setting:** BERT Base Size, Bigger Training Data, and/or More Training Steps | | | | | | | | | | | | |
| XLNet (Yang et al., 2019) | 110M | 86.8/- | 91.4 | 91.7 | 94.7 | 60.2 | 74.0 | 88.2 | 89.5 | 84.6 | 80.2 | – |
| RoBERTa (Liu et al., 2019) | 125M | 87.6/- | 91.9 | 92.8 | 94.8 | 63.6 | 78.7 | 90.2 | 91.2 | 86.4 | 80.5 | 83.7 |
| UniLM V2 (Bao et al., 2020) | 110M | 88.5/- | 91.7 | 93.5 | 95.1 | 65.2 | 81.3 | **91.8** | 91.0 | 87.1 | 83.3 | 86.1 |
| DeBERTa (He et al., 2021) | 134M | 88.8/88.5 | – | – | – | – | – | – | – | – | 83.1 | 86.2 |
| CLEAR (Wu et al., 2020) | 110M | 86.7/- | 90.0 | 92.9 | 94.5 | 64.3 | 78.3 | 89.2 | 89.8 | 85.7 | – | – |
| COCO-LM (Meng et al., 2021) | 134M | 90.2/90.0 | 92.2 | 94.2 | 94.6 | 67.3 | **87.4** | 91.2 | 91.8 | 88.6 | **85.4** | **88.1** |
| AMOS | 134M | **90.5/90.4** | **92.4** | **94.4** | **95.5** | **71.8** | 86.6 | 91.7 | **92.0** | **89.4** | 85.0 | 87.9 |

protocols: Single task learning with vanilla fine-tuning and reporting the median of five random seeds in GLUE and SQuAD. Please refer to Appendix A for more details.

**Baselines.** We compare with various pretrained models in both settings. All numbers are from reported results in recent research (more details in Appendix J).

**Implementation Details.** Our implementation builds upon the open-source implementation of fairseq Ott et al. (2019). Standard hyperparameters in pretraining and fine-tuning are used. More details are listed in Appendix B.

# 5 EVALUATION RESULTS

We conduct three groups of experiments to evaluate the performance of AMOS, the effect of leveraging signals from multiple generators, and the influence of its adversarial training. We also provide some case studies on the constructed pretraining sequences in AMOS.

## 5.1 RESULTS AND ABLATIONS ON GLUE AND SQUAD

**GLUE and SQuAD Results.** Table 1 lists the single-task fine-tuning performance of AMOS and notable baselines that are pretrained under the standard *base* and *base++* setting. AMOS outperforms all previous state-of-the-art pretraining methods on the overall GLUE score and SQuAD. AMOS's improvements are robust, often of large margins, and achieves the new state-of-the-art on multiple tasks under this standard pretraining/fine-tuning setup.

Table 2 presents the ablation studies of AMOS variants on MNLI and SQuAD, the two most stable and representative downstream evaluations (Refer to Appendix E for the full results). We organize the ablations into four subgroups to study the effectiveness of *Curriculum Learning*, *Adversarial Training*, *Multi-Layer MLM*, and *Training Signal Diversity* in AMOS.

**Effect of Curriculum Setup.** Disabling the automatic mixture weights learning in AMOS leads to downgraded downstream performance: Randomly picking the MLM head from the same set of generator layers (*w. random layer*) reduces accuracy on all metrics. It increases the diversity of pretraining signals but does not configure then as effectively. Manually configuring the order of different sets of pretraining signals by manually switching from shallower to deeper layers (*i.e.*, 4 to 6 to 8) during pretraining (*w. layer switch*) does not lead to better results than random layer selection. Note that in these manual configurations we manually switch the signals at $1/3$ and $2/3$ of the total

Table 2: Ablations on MNLI/SQuAD 2.0 dev sets that remove (-), add (+) or switch (w.) one component. Values are differences (in absolute points) from AMOS$_{Base}$.

| Group | Method | MNLI (m/mm) | SQuAD 2.0 EM/F1 |
|---|---|---|---|
| | AMOS$_{Base}$ | 88.9/88.7 | 84.2/87.2 |
| Curriculum Setup | w. random layer | -0.3/-0.4 | -0.6/-0.6 |
| | w. layer switch | -0.3/-0.3 | -0.9/-1.0 |
| Adversarial Setup | - adv. train | -0.2/-0.2 | -0.3/-0.4 |
| | + adv. MLM | -0.3/0.0 | 0.0/0.0 |
| Multi-MLM Setup | - stop grad. | -0.5/-0.1 | -0.6/-0.6 |
| | w. separate MLM gen. | -0.1/0.0 | -0.8/-0.7 |
| Backbone (No Multi-MLM No Adv. Train) | 4-layer gen. | -0.5/-0.5 | -1.1/-1.2 |
| | 6-layer gen. | -0.3/-0.4 | -1.1/-1.1 |
| | 8-layer gen. | -0.6/-0.6 | -0.9/-0.9 |
| | 12-layer gen. | -1.1/-1.2 | -1.8/-1.7 |

Table 3: Edge probing results using different MLM layers from the AMOS generator. Tasks are ordered based on suggested semantic depths in Tenney et al. (2019a).

| Tasks | layer 4 | layer 6 | layer 8 |
|---|---|---|---|
| POS | 92.6 | **93.4** | 91.2 |
| Consts. | 69.7 | **73.2** | 73.0 |
| Deps. | 86.4 | 85.1 | **88.3** |
| Entities | 91.7 | **93.9** | **93.9** |
| SRL | 76.9 | 74.2 | **79.2** |
| Coref. | 77.6 | 75.5 | **77.9** |
| SPR2 | 79.4 | 79.0 | **79.9** |
| Relations | 72.2 | **76.3** | 73.9 |

pretraining steps. We have experimented switching at different steps of pretraining but do not observe significantly better results. Manual trials are tedious and expensive in pretraining and underperform automatically learned mixture over multiple training signals.

**Effect of Adversarial Setup.** Not performing adversarial learning (- *adv. train*) to learn mixture weights (*i.e.*, always use unweighted average signals) hurts the model performance. However, note that this ablation still benefits from the curriculum learning effect as the generator gradually learns better. In addition, we also try to backpropagate the adversarial gradient to update the MLM embeddings (+ *adv. MLM*). Specifically, in Eqn. (3), both $\gamma_i^{(d)}$ and $h_i^{(d)}$ are updated by the reversed gradient from the discriminator. Our observation is that this makes the training unstable, perhaps because it hinders the MLM task of the auxiliary model. We have to use a very small gradient multiplier (*e.g.*, 0.1) when updating the MLM embeddings with the discriminator's backpropagated gradient, which has minimal effects on the model.

**Effect of Multi-Layer Generator Setup.** We also experimented with several configurations of the multi-layer generators: *-stop grad.* does not stop the gradient at the 4th and 6th layers in the generator of AMOS; *w. separate MLM gen.* trains three separate generators of 4, 6 and 8 layers jointly with the discriminator. Both configurations result in reduced performance of the discriminator and we keep the simpler setup as in AMOS. Additional experiments on using different numbers of MLM heads can be found in Appendix F.

**Effect of Diverse Training Signals.** The last ablation compares with using a single generator in AMOS and without any learned mixtures. Using a 4/6/8-layer generator yields worse results than AMOS and previous ablations with the multi-MLM generator, especially on SQuAD. The 12-layer generator is too strong and makes the pretrained discriminator significantly worse. It is simpler and more effective to grant the model access to a broader set of training signals and automatically learn to leverage them.

**Pretraining Efficiency.** Pretraining efficiency study of AMOS and comparison with previous models can be found in Appendix C.

## 5.2 Diverse Pretraining Signals from Different Generators

To study how generators of different depths provide diverse pretraining signals, we conduct probing experiments on eight NLP tasks covering different linguistic aspects, following (Tenney et al., 2019a;b): part-of-speech (POS) (Weischedel et al., 2013), constituents (Consts.), dependencies (Deps.), entities, semantic role labeling (SRL), coreference (Coref.), semantic proto-roles (SPR2 (Rudinger et al., 2018)), and relation classification (SemEval (Hendrickx et al., 2010)). The major difference between our setting and Tenney et al. (2019a;b) is that we do not combine embeddings from multiple layers but directly use the embedding from each MLM layer of AMOS generator as the (frozen) feature to a trainable linear classifier, as we are interested in what information is captured by each MLM layer.

As shown in Table 3, different MLM layers in AMOS generator indeed are good at different tasks–The 6th layer in the generator has the best performance on POS, constituent labeling, entity recognition

and relation classification, while the 8th layer performs the best on the other tasks. This demonstrates that using different layers in the multi-layer MLM generator is helpful for creating training signals of different levels of difficulty and also emphasizing different linguistic aspects. Note that although the 4th layer has worse performance than deeper layers across all tasks, it is still useful for providing discriminator pretraining signals, because the discriminator needs to learn from the "mistakes" made by the generator not capturing certain language semantics. Some concrete case studies of replaced tokens generated by different generator layers can be found in Appendix I.

## 5.3   Effects of Adversarial Training

To better understand the adversarial training dynamics, we plot the mixture weights (averaged over all masked positions) assigned to the 4th/6th/8th layer MLM during pretraining in Figure 3a. In the later pretraining stage, the 8th layer gradually gains more weights with the weights of the other two layers decreasing, showing that the generator indeed tries to create more challenging training signals for the discriminator. We also compare the training process of AMOS with several generator ablations: (1) A 4-layer generator;

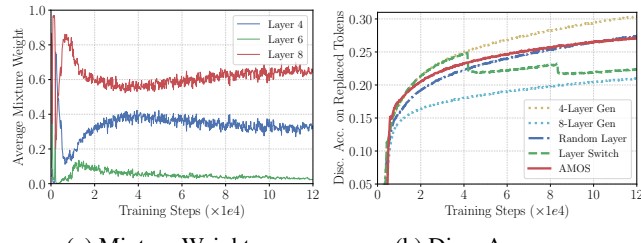

(a) Mixture Weights.          (b) Disc. Accuracy.

Figure 3: (a) The average mixture weights of the 4th/6th/8th generator layers on all masked positions during pretraining. (b) The discriminator accuracy on replaced tokens during pretraining under different curriculum learning setups.

(2) An 8-layer generator; (3) A 4/6/8-multi-layer MLM generator with three layers randomly picked for generating MLM replacements (Random Layer); (4) Switching from shallow layers to deeper layers (*i.e.*, 4 to 6 to 8) at $1/3$ and $2/3$ of the pretraining steps (Layer Switch). We plot the discriminator accuracy (averaged on replaced tokens) in Figure 3b. AMOS creates an intuitive learning curriculum that starts simpler than random layer mixing and becomes more challenging as pretraining goes on. Making "hard" switches suddenly changes the task difficulty and may disrupt training.

## 6   Conclusions and Future Work

In this paper we present AMOS, a new strategy for pretraining ELECTRA-style text encoders using an adversarial mixture of multiple training signal generators. AMOS constructs the corrupted text sequences by attaching multiple MLM heads to a deeper generator and sampling replaced tokens from their mixed outputs. The weights of the mixtures are learned to maximize the training signals difficulty for the discriminator, by backpropagating the reversed gradient from the discriminator through Gumbel-Softmax. This upgrades the ELECTRA-style pretraining framework with an automatically learned curriculum that composes more diverse pretraining signals.

Our experiments on the GLUE and SQuAD benchmarks demonstrate the empirical advantages of AMOS. Under the standard BERT$_{Base}$ and RoBERTa$_{Base++}$ pretraining settings, the same Transformer network pretrained with AMOS achieves the new state-of-the-art in nearly all evaluation metrics, with around 1 point gains on GLUE score and SQuAD accuracy. Our studies and analyses further confirm the source of AMOS's effectiveness: The more diverse training signals from multiple generators and our adversarial learning design to effectively utilize them throughout pretraining.

Our observations can be viewed as another progress of "data-centric" AI—Effectively constructed training signals from data can lead to significant empirical improvements without changing the model itself. Future work along this direction includes but not limited to: More studies to understand the role of training signals in language model pretraining, explorations of a broader set of training signal sources, and better strategies to leverage different information sources in pretraining.

## REPRODUCIBILITY STATEMENT

We strive to facilitate the reproducibility of the reported results in this paper, by (1) using the same pretraining and fine-tuning setups with previous research and reporting the median results of multiple fine-tuning runs, (2) providing details about the datasets used in Appendix A, sources of baselines compared in Appendix J, and hyperparameters used in Appendix B, (3) reporting the exact pretraining hours of our base models in Appendix C, and (4) releasing our pretrained models. All experiments in this paper are conducted on $64$ A100 GPUs each with $40$GB memory size.

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

## A    DETAILS OF GLUE TASKS

Table 4: GLUE task statistics and information.

|        | Size  | Domain        | Task          | Metric(s)         |
|--------|-------|---------------|---------------|-------------------|
| MNLI   | 393K  | Misc.         | Inference     | Accuracy          |
| QQP    | 364K  | Social QA     | Similarity    | Accuracy/F1       |
| QNLI   | 108K  | Wikipedia     | QA/Inference  | Accuracy          |
| SST-2  | 67K   | Movie Reviews | Sentiment     | Accuracy          |
| CoLA   | 8.5K  | Misc.         | Acceptability | Matthews corr.    |
| RTE    | 2.5K  | Misc.         | Inference     | Accuracy          |
| MRPC   | 3.7K  | News          | Paraphrase    | Accuracy/F1       |
| STS-B  | 5.7K  | Misc.         | Similarity    | Pearson/Spearman. |

Below are detailed descriptions of the tasks included in the GLUE benchmark. The statistics can be found in Table 4.

**MNLI:** Multi-genre Natural Language Inference (Williams et al., 2018) has 393K training examples via crowdsourcing. The task is to predict whether a premise sentence entails, contradicts or is neutral to a given hypothesis sentence.

**QQP:** Question Pairs (Shankar et al., 2017) has 364K training examples from the Quora question-answering website. The task is to determine whether a given pair of questions asked are equivalent semantically.

**QNLI:** Question Natural Language Inference has 108K training examples collected from the Stanford Question Answering Dataset (SQuAD) (Rajpurkar et al., 2016). The task is to predict whether a given sentence includes the answer to a given question sentence.

**SST-2:** Stanford Sentiment Treebank (Socher et al., 2013) has 67K training examples obtained from movie reviews with manually-annotated sentiment scores. The tasks is to determine if the sentence contains positive or negative sentiment.

**CoLA:** Corpus of Linguistic Acceptability (Warstadt et al., 2019) has 8.5K training examples collected from books and journal articles on linguistic theory. The task is to determine whether a sentence is linguistically acceptable or not.

**RTE:** Recognizing Textual Entailment (Dagan et al., 2005; Haim et al., 2006; Giampiccolo et al., 2007; Bentivogli et al., 2009) has 2.5K training examples from textual entailment challenges. The task is to predict whether a premise sentence entails a hypothesis sentence or not.

**MRPC:** Microsoft Research Paraphrase Corpus (Dolan & Brockett, 2005) contains 3.7K training examples from online news sources. The task is to predict whether two sentences are equivalent semantically.

**STS-B:** Semantic Textual Similarity (Cer et al., 2017) contains 5.8K training examples collected from multiple sources with human annotations of sentence pair semantic similarity. The task is to predict how semantically similar two sentences are (with a 1 to 5 scoring scale).

## B    HYPERPARAMETER SETTINGS

We use the default/standard values for most hyperparameters for pretraining: The generator pretraining uses the standard $15\%$ masking ratio. The temperature for Gumbel-Softmax is $\tau = 0.3$. The

Table 5: Hyperparameters used in pretraining.

| Parameters | *base* | *base++* |
|---|---|---|
| Max Steps | 125K | 1.95M |
| Peak Learning Rate | 5e-4 | 2e-4 |
| Batch Size | 2048 | 2048 |
| Warm-up Steps | 10K | 10K |
| Sequence Length | 512 | 512 |
| Adam $\epsilon$ | 1e-6 | 1e-6 |
| Adam $(\beta_1, \beta_2)$ | (0.9, 0.98) | (0.9, 0.98) |
| Clip Norm | 2.0 | 2.0 |
| Dropout | 0.1 | 0.1 |

Table 6: Hyperparameter ranges searched for fine-tuning.

| Parameters | GLUE Fine-tuning | SQuAD Fine-tuning |
|---|---|---|
| Max Epochs | {2, 3, 5, 10} | {2, 3} |
| Peak Learning Rate | {1e-5, 2e-5, 3e-5, 4e-5, 5e-5} | {2e-5, 3e-5, 4e-5, 5e-5} |
| Batch Size | {16, 32} | {16, 32} |
| Warm-up Proportion | {6%, 10%} | {6%, 10%} |
| Sequence Length | 512 | 512 |
| Adam $\epsilon$ | 1e-6 | 1e-6 |
| Adam $(\beta_1, \beta_2)$ | (0.9, 0.98) | (0.9, 0.98) |
| Clip Norm | - | - |
| Dropout | 0.1 | 0.1 |

discriminator loss weight $\lambda = 50$ since the loss of the binary classification task is much lower than the MLM task, which is a $30,000$-way classification task. The token embeddings are shared between the generator Transformer and the discriminator Transformer. Other hyperparameters used in pretraining and fine-tuning are reported in Tables 5 and 6, respectively.

The same (or equivalent) set of hyperparameters for pretraining and fine-tuning are used for all compared methods. The reported downstream task results on GLUE/SQuAD are the median of five runs with the same set of random seeds.

## C  PRETRAINING EFFICIENCY

We compare the pretraining efficiency of AMOS with COCO-LM (Meng et al., 2021), ELEC-TRA (Clark et al., 2020) and RoBERTa (Liu et al., 2019) under exactly the same computation environment for *base* model training. We show the MNLI-(m/mm) development set accuracy (via standard fine-tuning) of AMOS checkpoints trained for different GPU hours in Figure 4. For fair comparisons, we train all compared models using the same codebase, pretraining configuration, and computing environments. While AMOS takes longer to train than RoBERTa and ELECTRA, it matches their final MNLI performance with significantly fewer pretraining hours and achieves much better performance upon convergence. For example, it achieves RoBERTa's MNLI accuracy with two hours of pretraining

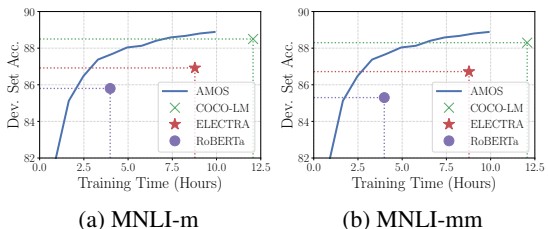

(a) MNLI-m   (b) MNLI-mm

Figure 4: AMOS$_{\text{Base}}$ accuracy on MNLI Dev. sets (y-axes) at different pretraining hours on 64 A100 (40 GB Memory) GPUs. The final training hours and accuracy of COCO-LM, ELECTRA and RoBERTa (trained under the exact same settings and computing environments) are also shown.

and outperforms ELECTRA in three hours, more than a $50\%$ reduction in pretraining time. It also reaches the accuracy of COCO-LM, the recent state-of-the-art in both pretraining accuracy and efficiency, only using $60\%$ of COCO-LM's pretraining hours. This demonstrates the advantage of the adversarial curriculum learning of AMOS.

Table 7: GLUE test set scores obtained from the GLUE leaderboard. We follow the standard in recent research to construct the test predictions: Searching for best hyperparameters with ten random seeds on each task individually and using the best development set model on testing data. All results are from vanilla single-task fine-tuning (no ensemble, task-specific tricks, etc.).

| Model | Params | MNLI-(m/mm) | QQP | QNLI | SST-2 | CoLA | RTE | MRPC | STS-B | AVG |
|---|---|---|---|---|---|---|---|---|---|---|
| **Base Setting:** | | | | | | | | | | |
| BERT | 110M | 84.6/83.4 | 89.2 | 90.5 | 93.5 | 52.1 | 66.4 | 84.8 | 85.8 | 80.8 |
| ELECTRA | 110M | 85.8/– | 89.1 | 92.7 | 93.4 | 59.7 | 73.1 | 86.7 | 87.7 | 83.5 |
| COCO-LM | 110M | 88.3/**88.1** | 89.9 | 93.3 | 94.9 | 61.9 | **81.5** | 87.8 | 88.6 | 85.8 |
| AMOS | 110M | **88.9/88.1** | **90.0** | **93.6** | **95.3** | **68.7** | 81.1 | **88.5** | **90.2** | **87.0** |
| **Base++ Setting:** | | | | | | | | | | |
| ELECTRA | 110M | 88.5/88.0 | 89.5 | 93.1 | 96.0 | 64.6 | 75.2 | 88.1 | 90.2 | 85.6 |
| COCO-LM | 134M | 89.8/89.3 | 89.8 | 94.2 | 95.6 | 68.6 | 82.3 | 88.5 | 90.3 | 87.4 |
| AMOS | 134M | **90.4/89.9** | **90.2** | **94.6** | **96.8** | **69.2** | **83.6** | **88.9** | **91.3** | **88.1** |

Table 8: Ablations on the development sets of all GLUE tasks and SQuAD 2.0 that eliminate (-), add (+) or switch (w.) one component. We show the median and standard deviation (as subscripts) of five random seeds on each task. The results are extensions of Table 2.

| Group | Method | GLUE Single Task | | | | | | | | SQuAD 2.0 | |
|---|---|---|---|---|---|---|---|---|---|---|---|
| | | MNLI-(m/mm) | QQP | QNLI | SST-2 | CoLA | RTE | MRPC | STS-B | EM | F1 |
| | $AMOS_{Base}$ | **88.9**$_{0.1}$/**88.7**$_{0.1}$ | **92.3**$_{0.0}$ | 93.6$_{0.1}$ | **94.2**$_{0.2}$ | 70.7$_{1.5}$ | 86.6$_{1.4}$ | 90.9$_{0.4}$ | 91.6$_{0.1}$ | **84.2**$_{0.2}$ | **87.2**$_{0.3}$ |
| **Curriculum Setup** | w. random layer | 88.6$_{0.1}$/88.3$_{0.1}$ | 92.2$_{0.1}$ | 93.2$_{0.2}$ | 93.9$_{0.2}$ | 70.2$_{1.6}$ | 84.8$_{1.0}$ | **91.4**$_{0.7}$ | **91.8**$_{0.2}$ | 83.6$_{0.2}$ | 86.6$_{0.2}$ |
| | w. layer switch | 88.6$_{0.1}$/88.4$_{0.1}$ | 92.2$_{0.1}$ | 93.0$_{0.1}$ | 94.0$_{0.3}$ | 70.2$_{1.0}$ | 85.6$_{0.9}$ | 90.9$_{1.0}$ | 91.6$_{0.1}$ | 83.3$_{0.3}$ | 86.2$_{0.1}$ |
| **Adversarial Setup** | - adv. train | 88.7$_{0.1}$/88.5$_{0.1}$ | **92.3**$_{0.1}$ | 93.2$_{0.1}$ | **94.2**$_{0.2}$ | 71.3$_{0.9}$ | **87.0**$_{1.1}$ | 90.9$_{0.9}$ | 91.5$_{0.1}$ | 83.9$_{0.1}$ | 86.8$_{0.2}$ |
| | + adv. MLM | 88.6$_{0.1}$/**88.7**$_{0.2}$ | 92.2$_{0.1}$ | 93.5$_{0.2}$ | 93.8$_{0.3}$ | 71.3$_{1.0}$ | 85.9$_{1.2}$ | 91.4$_{1.0}$ | **91.8**$_{0.1}$ | **84.2**$_{0.2}$ | **87.2**$_{0.1}$ |
| | w. final mix weights | 88.4$_{0.1}$/88.0$_{0.1}$ | 92.2$_{0.1}$ | 93.0$_{0.2}$ | 93.6$_{0.3}$ | 70.9$_{0.8}$ | 83.0$_{1.0}$ | 90.9$_{0.5}$ | 91.3$_{0.2}$ | 83.5$_{0.1}$ | 86.5$_{0.1}$ |
| **Multi-MLM Setup** | - stop grad. | 88.4$_{0.1}$/88.6$_{0.1}$ | **92.3**$_{0.1}$ | **93.9**$_{0.2}$ | 93.9$_{0.4}$ | 71.1$_{1.1}$ | **87.0**$_{2.0}$ | 91.2$_{0.7}$ | 91.6$_{0.1}$ | 83.6$_{0.2}$ | 86.6$_{0.2}$ |
| | w. separate MLM gen. | 88.8$_{0.1}$/**88.7**$_{0.2}$ | **92.3**$_{0.1}$ | 93.2$_{0.1}$ | 94.0$_{0.2}$ | **72.1**$_{0.8}$ | 85.2$_{1.2}$ | 90.7$_{0.6}$ | 91.6$_{0.1}$ | 83.4$_{0.3}$ | 86.5$_{0.3}$ |
| | w. three 4-layer gen. | 88.6$_{0.1}$/88.4$_{0.3}$ | 91.9$_{0.0}$ | 92.8$_{0.2}$ | 93.9$_{1.0}$ | 69.7$_{1.3}$ | 83.0$_{0.9}$ | 90.9$_{0.1}$ | 91.2$_{0.2}$ | 83.0$_{0.3}$ | 85.9$_{0.2}$ |
| **Backbone (No Multi-MLM No Adv. Train)** | 4-layer gen. | 88.4$_{0.1}$/88.2$_{0.0}$ | 92.2$_{0.0}$ | 93.1$_{0.2}$ | **94.2**$_{0.5}$ | 69.2$_{1.1}$ | 84.8$_{1.0}$ | 91.2$_{0.4}$ | 91.3$_{0.1}$ | 83.1$_{0.3}$ | 86.0$_{0.2}$ |
| | 6-layer gen. | 88.6$_{0.1}$/88.3$_{0.1}$ | 92.2$_{0.1}$ | 93.2$_{0.1}$ | 93.2$_{0.5}$ | 69.2$_{1.7}$ | 83.4$_{1.1}$ | 90.2$_{0.7}$ | 91.4$_{0.2}$ | 83.2$_{0.2}$ | 86.1$_{0.2}$ |
| | 8-layer gen. | 88.3$_{0.2}$/88.1$_{0.1}$ | 92.1$_{0.1}$ | 93.0$_{0.2}$ | 93.5$_{0.5}$ | 70.0$_{1.2}$ | 85.2$_{1.0}$ | 90.7$_{1.0}$ | 91.2$_{0.1}$ | 83.3$_{0.3}$ | 86.3$_{0.3}$ |
| | 12-layer gen. | 87.8$_{0.2}$/87.5$_{0.2}$ | 92.1$_{0.1}$ | 92.7$_{0.1}$ | 92.7$_{0.2}$ | 69.3$_{0.4}$ | 81.6$_{0.5}$ | 90.0$_{0.7}$ | 91.4$_{0.2}$ | 82.4$_{0.2}$ | 85.5$_{0.2}$ |

## D GLUE TEST SET RESULTS

We show the GLUE test set scores obtained via private submissions to the GLUE leaderboard in Table 7. The baseline results are directly retrieved from the leaderboard or from their original papers. We use standard single-task fine-tuning to more directly reflect the improvements from pretrained models. The advantage of AMOS over strong baselines holds on the test set: Under both *base* and *base++* settings, AMOS outperforms ELECTRA (Clark et al., 2020) and COCO-LM (Meng et al., 2021) on almost every task. Still, we would like to note that smaller tasks such as CoLA, RTE, and MRPC are not stable and leaderboard runs often use more sophisticated fine-tuning method to achieve better performance, for example, continuing training from the checkpoints fine-tuned on MNLI.

## E MORE DETAILED ABLATION RESULTS

We show more detailed ablation results in Table 8 which extends Table 2 by including results from all GLUE tasks and showing the standard deviations. We also include two new ablations: *w. final mix weights* pretrains the discriminator all the way with the fixed mixture weights learned by AMOS at convergence; *w. three 4-layer gen.* trains the discriminator with mixtures over three 4-layer generators instead of the multi-layer MLM generator in AMOS.

AMOS has better performance than all other ablation versions on most large tasks (MNLI, QQP, SST-2 and SQuAD) which are considered more stable and reliable indicators of model effectiveness. Small tasks (CoLA, RTE, MRPC) have much higher variance than larger tasks, and usually require intermediate task training to yield stable results (*i.e.*, starting from checkpoints that are fine-tuned on MNLI (Clark et al., 2020; He et al., 2021; Liu et al., 2019)).

Table 9: Performance study with different numbers of MLM heads used in the generator. We show the median and standard deviation (as subscripts) of five random seeds on each task.

| MLM Layers | GLUE Single Task | | | | | | | | SQuAD 2.0 | |
| --- | --- | --- | --- | --- | --- | --- | --- | --- | --- | --- |
| | MNLI-(m/mm) | QQP | QNLI | SST-2 | CoLA | RTE | MRPC | STS-B | EM | F1 |
| On (4,6,8) (AMOS) | $\mathbf{88.9}_{0.1}/\mathbf{88.7}_{0.1}$ | $\mathbf{92.3}_{0.0}$ | $\mathbf{93.6}_{0.1}$ | $94.2_{0.2}$ | $70.7_{1.5}$ | $\mathbf{86.6}_{1.4}$ | $90.9_{0.4}$ | $\mathbf{91.6}_{0.1}$ | $\mathbf{84.2}_{0.2}$ | $\mathbf{87.2}_{0.3}$ |
| On (4,8) | $88.6_{0.2}/88.4_{0.2}$ | $\mathbf{92.3}_{0.1}$ | $93.4_{0.1}$ | $\mathbf{94.4}_{0.2}$ | $68.9_{1.2}$ | $85.9_{0.7}$ | $\mathbf{91.4}_{0.9}$ | $91.5_{0.1}$ | $83.8_{0.2}$ | $86.7_{0.2}$ |
| On (2,4,6,8) | $88.4_{0.1}/88.3_{0.2}$ | $92.2_{0.1}$ | $93.5_{0.2}$ | $93.8_{0.3}$ | $69.1_{0.7}$ | $84.5_{1.5}$ | $90.9_{0.4}$ | $91.3_{0.2}$ | $83.8_{0.2}$ | $86.8_{0.2}$ |
| On all 8 | $88.2_{0.1}/88.0_{0.2}$ | $92.2_{0.1}$ | $93.1_{0.1}$ | $94.0_{0.4}$ | $\mathbf{71.5}_{1.4}$ | $84.8_{1.0}$ | $90.7_{1.0}$ | $\mathbf{91.6}_{0.2}$ | $83.1_{0.2}$ | $86.0_{0.1}$ |

The instability of the GLUE small tasks is a widely-observed artifact in pretraining research. It is standard (and recommended) practice to *not* rely on these small unstable tasks for ablation studies. For example, to ensure a correct understanding of research progress, previous studies including BERT (Devlin et al., 2019), RoBERTa (Liu et al., 2019), ELECTRA (Clark et al., 2020), and DeBERTa (He et al., 2021) all perform ablation studies on large GLUE tasks like MNLI. We include the ablation results on small GLUE tasks in Table 8 *only for reference*.

Furthermore, we show the standard derivations of the same pretrained checkpoint when fine-tuned on the corresponding tasks with different random seeds. They further confirm that these smaller tasks have large variance in the fine-tuning stage. Observing the variance of the pretraining stage (*i.e.*, changes in model performance when pretrained with different random seeds) requires running the costly pretraining multiple times which is often infeasible. We would like to refer to a recent study (Sellam et al., 2022) which reveals that these small GLUE tasks are very unstable, and the same model pretrained with different random initialization can have $2-5$ points difference in performance upon fine-tuning on these small tasks, while in comparison, observations on MNLI are more reliable. Since in this paper we only perform single-task fine-tuning without any intermediate task training, we mainly rely on large task results for evaluating the effectiveness of different model components. The GLUE average score is more of a convenient reference point as used in the pretraining research community.

## F    PERFORMANCE STUDY WITH DIFFERENT NUMBERS OF MLM HEADS

In this experiment, we study the performance of AMOS with different numbers of generator MLM heads $K$. In addition to the original AMOS which uses three MLM heads, we also show in Table 9 the downstream task performance when two MLM heads (at the 4th and 8th layers), four MLM heads (at the 2nd, 4th, 6th and 8th layers) and eight MLM heads (at each layer) are used. We note that increasing MLM heads does not necessarily lead to better performance, since having more MLM heads means that each MLM block (partitioned by the stop gradient operators at each MLM layer) will become shallower with weaker MLM learning capacity. We have also tried inserting the same amount of MLM heads at different layers of the generator, but it does not improve the results.

## G    DISCUSSIONS ON USING GUMBEL-SOFTMAX FOR GRADIENT ESTIMATION

In this work, we use Gumbel-Softmax to enable gradient approximation of the non-differentiable sampling operation so that we are able to use the gradient backpropagated from the discriminator to train the mixture weights. There are other possible approaches for discriminator gradient estimation, including REINFORCE (Yu et al., 2017) which has been explored in the ablation studies of ELECTRA (Clark et al., 2020), or directly operating in the hidden states of the generator instead of in its discrete output space which has been studied in adversarial learning based text generation research (Subramanian et al., 2018; Zhang et al., 2017; Zhao et al., 2018). We would like to note that as our first study on adversarial curriculum for language model pretraining, we prefer a simple framework and standard techniques to demonstrate that such a new direction is promising. We believe that exploring more sophisticated and advanced realizations for specific model components in our AMOS framework (*e.g.*, using better gradient estimators than Gumbel-Softmax) will be an interesting future work direction.

Table 11: Examples of replaced tokens by different-sized MLM generators. Underlined words are masked out; the replaced tokens by 4/6/8-layer generators are marked in different colors.

---

**Example 1.** "Here, let me show you that animal." He pointed up to (at/at/into) the canopy; hanging from a white branch was (of/stood/stood) a pale, hairless creature (shape/plant/animal). It was bilaterally symmetrical, with two pairs of tightly (long/firmly/strongly) folded limbs, but did not appear to have any discernible head.

---

**Example 2.** In 2012, about 3,000 (6/1,800/1,800) villagers remained in Rammun, while there are about 7,000 in the Palestinian diaspora, chiefly in the United States. Many in the diaspora have second homes in the (this/the/their) village. These homes have been troubled by burglaries, therefore some owners have organised night-watches (things/boxes/searches).

---

## H PROBING EXPERIMENTS WITH DIFFERENT SETUPS

In addition to the probing experiments conducted in Section 5.2, we also show the results under a different setup: Instead of testing the linguistic information captured by different MLM layers in AMOS generator, we conduct the same tests on separate generators with different depths (4 layers, 6 layers and 8 layers). The probing test results on eight tasks are shown in Table 10. Similar to the results in Table 3, generators of different depths are good at capturing different types of linguistic information. Therefore, it is also possible to compose diverse training signals with multiple separate generators with different numbers of layers. Such findings may be useful for future studies that explore using multiple different-sized auxiliary models to provide training signals emphasizing different linguistic aspects.

Table 10: Edge probing results of (separate) generators with different numbers of Transformer layers.

| Tasks | 4 layer | 6 layer | 8 layer |
|---|---|---|---|
| POS | 93.7 | **94.7** | 94.3 |
| Consts. | 75.0 | 76.2 | **76.3** |
| Deps. | 88.8 | 89.4 | **89.5** |
| Entities | 93.6 | 94.8 | **94.9** |
| SRL | 80.9 | 81.6 | **82.3** |
| Coref. | 80.0 | **81.7** | **81.7** |
| SPR2 | 79.8 | **81.3** | 80.2 |
| Relations | 74.5 | **75.8** | 75.0 |

## I CASE STUDY

Table 11 provides concrete cases of replaced tokens generated by different generator layers. The "mistakes" made by different generator layers have different levels of difficulty to be detected–The 4th layer MLM sometimes makes simple syntactic mistakes while the replaced tokens given by the 6th/8th layer are mostly plausible and need to be distinguished based on a deep understanding of the full contexts. This intuitively confirms our motivation that using a generator of multiple MLM heads can provide diverse pretraining signals to compose a more effective learning curriculum.

## J THE ORIGINS OF REPORTED BASELINE SCORES

The baseline results reported in Table 1 are obtained from the corresponding papers except the following: BERT/RoBERTa from Bao et al. (2020), ELECTRA from Meng et al. (2021), XLNet *base++* from Bao et al. (2020). When there are different reported scores for the same method, we use the highest of them in our comparisons.

## K SOCIETAL IMPACT

There have been concerns about the extensive costs of computing resource required by pretraining language models. The concerns include whether it is worthwhile to spend such enormous amount of resources in pretraining, and also whether the demanding resource requirements have posed a barrier for most institutions to conduct pretraining research, thus slowing down the overall scientific development.

One major motivation of this work is to more efficiently pretrain language models, including (1) to achieve better model effectiveness with fixed computing resource, and (2) to achieve the same

downstream task accuracy with fewer pretraining computes. As shown in our experimental results, we are able to obtain certain successes in both fronts. We demonstrate that one can achieve the same pretraining effectiveness of RoBERTa$_{\text{Base}}$ in $128$ GPU hours on A100 using AMOS (two hours on $64$ A100 GPUs), which is quite affordable for many research institutions. We hope our observation will inspire more future studies in efficient pretraining and also enable conducting pretraining research in more accessible computing environments.

In addition, we also mainly focus on the base-sized models in our exploratory research which are less costly than large or extra large models, while also being the most commonly used model configuration in the community. We release our pretrained model checkpoints to the community as well.

