# OpenReview forum: "Pretraining Text Encoders with Adversarial Mixture of Training Signal Generators"
_ICLR.cc/2022/Conference — ICLR 2022 Poster_

### Official Review · Reviewer_oMcQ · 2021-11-02

**Correctness:** 3
**Technical Novelty And Significance:** 3
**Empirical Novelty And Significance:** 3
**Recommendation:** 8
**Confidence:** 4

**Main Review:**

Strengths:
1) The paper is very well written and easy to follow. The paper is well-placed w.r.t recent developments in pre-training by presenting similarities and differences with related work.

2) The idea of using a committee of generators to create the replaced token for training the discriminator, so as to provide a diverse signal for learning different complexity levels of RTD is natural and logical (this also can be linked with curriculum learning). The neat trick utilized for modelling AMOS is to create multiple generators from the same single monolithic generator by means of individual MLM heads at different layer depths which : (i) reuses computation and is efficient, and (ii) exploits underlying representations learned by lower layers to make the deeper layer MLMs stronger.

3) The empirical results of the AMOS pretraining approach are strong. Despite several discussions in the NLP community on using GLUE as an evaluation benchmark, a 1 absolute point improvement using a 12-layer base model is significant over the previously strong baseline of COCO-LM (which additionally uses the InfoNCE loss).

4) The ablation studies on MNLI and SQuAD help justify the modeling decisions made in the paper w.r.t the learnable weights, the use of stop-gradient between layers, not using the adversarial discriminator loss to train the learnable weights, etc.


Weaknesses:
1) The ablation studies on MNLI and SQuAD are provided without standard deviation error bars. The improvements of AMOS over the ablated components is rather small, and thus it is important to present the error bar estimates to ensure the statistical significance of the results.

2) For Table 3, the experiments are performed using independent 4,6 and 8 layer generators to establish the point that the pretraining signals provided are diverse. A more appropriate experiment would have been to perform the pre-training of AMOS once, and then use the 4,6 and 8 -layer MLMs (having common transformer blocks) and then see whether this diversity in probing performance exists. The objective is to show that the 4,6 and 8 layer generator with the shared MLM have diverse probing performance on tasks.


3) While extensive, the experiments are limited to pretraining a 12-layer transformer model. Since the paper presents arguments such as the 12-layer generator being very strong, it is natural to question whether the improvements from AMOS will translate to larger architectures (24 layers) using the same mixture of multiple generators approach.


4) I think the paper should include a societal impact section and discuss the extensive compute resources that have been consumed for the pretraining runs and the ablation studies. Furthermore, the reproducibility statement is missing details of the compute infrastructure used : Number of GPUs, types of GPU/TPUs, etc.


Questions:
1) How was the \lambda in Equation-(4) chosen for the experiments? Was some form of cross-validation performed?

2) Instead of random masking, do you have any high level thoughts/intuitions on whether the AMOS approach will also provide empirical improvements with targetted masking approaches (For example: REALM that masks specific POS entities for retrieval-augmented MLM)?

3) While the different MLMs in AMOS have different number of layers, they all have the same structural transformer block, and thus are from the same "family" of generators. Since the goal is to increase the diversity and complexity of signal from the generator for improving the strength of the discriminator, it will be interesting to experiment with a different "family" of generators (based on LSTM, word-CNN, etc.). Any thoughts/comments on this?

4) The paper presents the AMOS multi-generator guided pretraining approach for a general k number of generators, but all the experiments are performed using k=3. While I understand that each pretraining run is computationally expensive, I was intrested to know if the authors had any intuitions/initial results from experiments by changing the number of MLM heads k?

**Summary Of The Paper:**

The paper presents a framework for pre-training transformer based LMs a la ELECTRA-Generator-Discriminator style by leveraging a mixture of generators and using signal from the discriminator to guide the learning of the generator. Specifically, the generator is modified to have multiple MLM heads specific to different layers (under the intuition that the deeper layers will act as a stronger generator), and the replacement predictions from each of the layers are combined using learnable weights as the input to the discriminator. To aid learning and ensure the different MLM heads of the generator to function as "independent generators", the gradient is not back-propagated across the entire generator, but only within individual blocks between the MLM heads. For back-propagating the discriminator loss for learning the mixture of generators, the Gumbel-Softmax trick is used to back-prop the loss to the weights (the discriminator-guided selection of different generators can be intuitively linked with GANs). Empirical evaluation is performed on GLUE and SQuAD, and the 12-layer AMOS pre-training (with both the standard base and base++ pretraining configurations) is able to outperform previous strong baselines like ELECTRA and COCO-LM by an average of 1+ absolute point. Ablation studies have been performed to show the empirical benefits of different components: the weights v/s random and simple->hard 3 model curriculum, back-propagating discriminator loss only to the weights and not to the MLM prediction heads, etc.


**Summary Of The Review:**

The paper makes a novel addition to the ELECTRA-style LM pretraining direction by using multiple generators to guide the learning of the discriminator, and in turn use the loss from the discriminator to enhance the generation capability of the generators. The idea is simple, easy to comprehend and evaluated on the standard benchmarks used for LM pretraining research. The results are strong and show a 1 point absolute improvement in average GLUE score over previous baselines.

The results of the ablation study are not very conclusive, and raise some doubts about whether certain modeling decisions are essential (and if they are making the AMOS modeling too complex). There are some weaknesses in the experiments to justify the need for different number of layers of generators, and concerns over generalization to different values of k and the size of the transformer architecture.

---

> ### Author Response · Authors · 2021-11-20
> **Response to Reviewer oMcQ**
>
> Thanks for your review! In our general response, we have listed the set of new experimental results and discussions added in the revised paper version. Below are more detailed discussions to your questions.
>
> Regarding the weaknesses:
> 1. **Evaluation with standard deviation error bars.** We have reported the median and standard deviation of all ablation runs on GLUE and SQuAD in the revision (please see Appendix F). MNLI and SQuAD are relatively stable which is also why we focus on them in our ablation study. We would like to note that the high-cost nature of pretraining makes it very challenging to study variances in pretraining compared to other research (so we reported variances brought out by the fine-tuning stage only). Our added studies are among one of the most detailed analyses in pretraining research studies. The added GLUE test results are another piece of evidence of our empirical contribution.
> 2. **Probing with 4,6 and 8 layer generator with the shared MLM.** We agree that your suggested experiment setup with shared MLM is more appropriate, and we have updated Table 3 with the new results. The diversity in linguistic aspects captured by MLMs at different depths still exists. We have moved the previous results to Appendix I for reference.
> 3. **Large model results.** We conducted all experiments on base-sized models because (1) training large models is considerably more costly than base ones, which is extremely energy-consuming and non-eco-friendly for the development of new methods; (2) base models are the standard settings for PLM research exploration and are also more widely used in downstream tasks in the community than large models; (3) PLM research explorations often start with base-sized models and many previous studies also only reported base-sized model results (_e.g._, UniLMv2 [1] and TUPE [2]). It is great if we can push all our techniques to large or even extra-large models in the future. That said, we hope our explorations and observations in the base/base++ model settings have sufficiently achieved our goal of demonstrating the potential of this new research direction (_i.e.,_ more fundamental ways to construct training signals in language model pretraining) and can inspire more future research.
> 4. **Include a societal impact section and report compute infrastructure for reproducibility.** We agree with your suggestions. We have added the societal impact section (please see Appendix J) and updated the reproducibility section in the revision.
>
> Regarding the questions:
> 1. **How was the $\lambda$ in Equation (4) set?** We set $\lambda=50$ following ELECTRA. We indeed tried a few different $\lambda$ values (_e.g._, 10, 20, 100) but they do not yield better performance so we keep it consistent with previous work.
> 2. **AMOS for targeted masking approaches.** AMOS and targeted masking are orthogonal methods for improving sample efficiency in pretraining: Targeted masking is usually concerned with "which tokens to mask in a sequence" whereas AMOS is focused on "given each mask position, what token to fill in". The goal of targeted masking as used in REALM and a later T5 version (referred to as the salient span masking) is generally to place emphasis on the semantic units that particularly benefit certain downstream tasks such as question answering. We expect that the two approaches will be able to work together in a system that carefully combines them.
> 3. **Using different "families" of generators.** We studied using Transformer-based MLMs as the generator because they are the most prevalent architecture in the current language model pretraining field. Using generators from other families (_e.g._, LSTMs, CNNs) may introduce different types of pretraining signals, but it will also make the training and tuning of the model even more complicated and costly---with fundamentally different architectures on the generator side, aligning generator training with discriminator training could be much more difficult (_e.g._, different model architectures may have different convergence rates with different optimal learning rates). It is also unclear whether non-Transformer models could provide signals that are difficult enough for the discriminator Transformer. As the first study to explore training signal diversity and adversarial curriculum in language model pretraining, we hope to keep our framework as simple and clear as possible. We agree that using generators from other families is an interesting future study direction.
> 4. **Changing the number of MLM heads.** We added this experimental study in Appendix G.
>
> References:
> [1] Hangbo Bao, Li Dong, Furu Wei, Wenhui Wang, Nan Yang, Xiaodong Liu, Yu Wang, Songhao Piao, Jianfeng Gao, Ming Zhou and Hsiao-Wuen Hon. "UniLMv2: Pseudo-Masked Language Models for Unified Language Model Pre-Training." ICML (2020).
> [2] Guolin Ke, Di He and Tie-Yan Liu. "Rethinking Positional Encoding in Language Pre-training." ICLR (2021).

---

> > ### Comment · Reviewer_oMcQ · 2021-11-24
> > **Thanks for the response (Increasing score to 8)**
> >
> > Thanks for the detailed response and considering my comments and suggestions in detail.
> >
> > * I agree that ablation studies on smaller GLUE datasets don't provide conclusive results due to high variance. I appreciate the detailed ablations with median and std deviations (I was just wondering why the mean was not used instead of median). While the difference is **small**, the results from Table 9 consistently show the AMOS-Base approach outperforming other ablated versions for MNLI and SQuAD.
> >
> > * Results in Appendix-G demonstrate the empirical benefits of the 3 MLM head configuration chosen by the authors.
> >
> > * Using different "families" of generators, combining AMOS with targeted MLM techniques and extending the approach to larger/extra-large  models are important future work avenues as an extension of this work.
> >
> > My major concerns and questions on the original draft of this paper have been mitigated by the author response and the updated version, and hence I am increasing my score to 8.

---

> > > ### Author Response · Authors · 2021-11-25
> > > **Response to Reviewer oMcQ Comments**
> > >
> > > Thanks for your positive feedback! We are glad that our response and update helped resolve your concerns.
> > > * We reported the median (instead of mean) of five fine-tuning runs because most previous studies (RoBERTa, ELECTRA, COCO-LM, etc.) reported median and we would like to make sure our results are directly comparable to them. The mean values are mostly the same as the median values (after rounding to one decimal place) in all our experiment results.
> > > * We completely agree with your visions of future work directions, including combining AMOS with targeted masking methods and scaling up the framework to train larger-sized models.

---

### Official Review · Reviewer_sJ99 · 2021-11-04

**Correctness:** 4
**Technical Novelty And Significance:** 3
**Empirical Novelty And Significance:** 3
**Recommendation:** 8
**Confidence:** 5

**Main Review:**

Strengths:

The paper is well written and easy to understand. Since it builds on the well known ELECTRA architecture, it is easy to identify the main contribution, which is adding more inductive learning curriculum to ELECTRA training. The authors achieved this by using multiple auxiliary MLMs and combining their outputs to generate the input to the discriminative text encoder. To improve training efficiency and further streamline the architecture, all the auxiliary MLMs are derived from the same base MLM with outputs of multiple layers used as its own sub-MLM realization. Finally, the mixture parameter is adversarially learned using the negative of the discriminator gradient. The ablation study is exhaustive providing empirical evidence of superiority of the proposed architecture.

Weaknesses:
There is a typo “date-centric” --> "data-centric"
The paper applied the MLM to only 3 layers (4/6/8) of the 8-layer generator. Do you have an intuition on the effect of increasing MLM heads on the discriminator performance on the downstream tasks? It was shown that using just one MLM head irrespective of the layer is worse, but is there an optimal number of of MLM heads for an N-layer generator network?

**Summary Of The Paper:**

This paper presents an adversarial learning framework for pre-training text encoders following ELECTRA-style architecture. The use of single auxiliary MLM broken into sub-MLMs and then combined together with a mixture parameter is novel. Authors spent great deal of effort on the ablation study to show that the proposed framework and design choices perform better than existing state-of-art methods and other alternatives on several downstream NLP tasks.

The paper is clearly written and easy follow. The conclusion of the inductive learning curriculum is well supported by experimental results.

**Summary Of The Review:**

I recommend this paper for publication given a satisfactory explanation of the raised weakness. Overall, I think the concepts explored in this paper are novel and will be of interest to the NLP community, and might spur new research in this direction.

---

> ### Author Response · Authors · 2021-11-20
> **Response to Reviewer sJ99**
>
> Thanks for your review! Besides your suggestions, we have also added a series of new experiments as listed in the general response.
>
> Regarding the weaknesses:
> 1. **Typo.** Thanks for pointing out the typo. We have corrected it in the revision.
> 2. **Increasing the number of MLM heads.** We added this experiment in Appendix G. With more MLM heads being added to more layers, especially those added to shallower layers, the generator's pretraining might get disturbed. An important future work direction is to further understand the characteristics of different ways to construct pretraining signals and their impact on the downstream performance.

---

> ### Author Response · Authors · 2021-11-27
> **Follow Up**
>
> Dear Reviewer sJ99,
>
> Thanks again for your review. We have answered your questions by updating our manuscript to incorporate new results (_e.g._, the performance study with more MLM heads). As the discussion period is ending soon, we would greatly appreciate it if you could provide feedback on whether you are satisfied with our response. We will be happy to address any remaining concerns.
>
> Best,
> Paper4429 Authors

---

### Official Review · Reviewer_CBi9 · 2021-11-07

**Correctness:** 3
**Technical Novelty And Significance:** 2
**Empirical Novelty And Significance:** 3
**Recommendation:** 6
**Confidence:** 3

**Main Review:**

The paper is clear written and relatively easy to follow. The paper idea is majorly from 3 aspects: curriculum learning, adversarial training via Gumble-softmax, layer wise understanding of pretrained models. These ideas are reasonable. The results show on GLUE dev and SQUAD 2.0 also looks significant.

There are two parts of the work I think needs further clarification. First, the author states that the model is based on mix of training signal generator, however, the work is actually using different layer mixture of signals. To make the statement solid, I think another experiment of independent generators with same layers as generators should be conducted. Also, the meaning of vector v in weight calculation is not clearly discussed. Second, the adversarial training does not include re-sample the masked tokens, which may be a strong signal in adverbial training.

Besides this, as this is a pretraining work. I would expect to see the performance on GLUE test set instead of just dev set which should make the results more convincing. Also, the training time/speed/performance comparison with ELECTRA may be better shown in the work.

**Summary Of The Paper:**

This work proposed a new framework AMOS to enhance ELECTRA-style pretraining. The new framework includes adversarial learning and curriculum learning. Instead of direct adverbial learning via Gumbel-softmax, the author proposed a mix-of-signals framework to mix the signal from different layer of the generator. The authors show the overall performance outperforms ELECTRA and more-recent SOTA COCO-LM by a reasonable margin.

**Summary Of The Review:**

Overall, I think this is a reasonable work by directly improving of the ELECTRA framework. The proposed ideas are clear, straightforward, and sound. The results can be further improvement.

---

> ### Author Response · Authors · 2021-11-20
> **Response to Reviewer CBi9**
>
> Thanks for your review!
>
> Regarding the points you raised:
> 1. **Using independent generators with same layers as generators.** We have conducted this experiment with three independent 4-layer generators as a new ablation and reported its performance in Appendix F (Table 9, w. three 4-layer gen.). Using multiple generators with the same depth does not perform well, confirming the benefits of having a diverse set of training signals of different levels of difficulty.
> 2. **The meaning of vector $\boldsymbol{v}$ in mixture weight calculation.** The computation of $\gamma_i^{(d)}$ involves Softmax normalization (so that the mixture weights sum to 1) as shown in Equation (3), so the vector $\boldsymbol{v}$ is essentially the weight in the Softmax layer, which can be conceptually interpreted as the hidden representation resulting in challenging replaced tokens that maximize discriminator loss---When the Transformer embedding $\boldsymbol{f}_i^{(d)}$ has a high dot product similarity with $\boldsymbol{v}$, the resulting mixture weight logit will be high, and the MLM head at that layer will play a more important role in generating the final replaced token.
> 3. **Including masked token re-sampling in the adversarial training**. We indeed have tried learning the masked positions in an adversarial manner similar to how the mixture weights are learned. Specifically, we learn a probability (between 0 and 1) for each token indicating how likely it should be masked so that the token is challenging for the discriminator. The mask probability is learned to maximize the discriminator loss via Gumbel-Softmax gradient estimation. We found that sampling mask positions according to such learned probability indeed results in higher discriminator loss compared to randomly sampling mask positions, but the discriminator downstream task performance does not improve. This observation is in line with the first exploration mentioned in ELECTRA paper's Appendix H: Negative Results.
> 4. **Performance on GLUE test set.** We have reported GLUE test set performance for _Base_ and _Base++_ settings in the revision (please see Appendix D). AMOS outperforms strong baselines on GLUE test set by significant margins as well.
> 5. **Training time and performance comparison with ELECTRA.** We have included the MNLI task performance _w.r.t._ training time of AMOS in the revision (please see Appendix E). AMOS matches the MNLI performance of RoBERTa and ELECTRA with about 50% of their pretraining time, demonstrating significantly higher pretraining sample efficiency. AMOS also outperforms COCO-LM with a shorter training time.

---

> > ### Comment · Reviewer_CBi9 · 2021-11-29
> > **Response to author**
> >
> > Thanks for the detailed explanation. All my questions have been addressed.

---

> ### Author Response · Authors · 2021-11-27
> **Follow Up**
>
> Dear Reviewer CBi9,
>
> Thanks again for your review. We have followed your suggestions by updating our manuscript to incorporate new results (_e.g._, the new ablation of using independent same-layer generators, GLUE test set performance, and training time/performance analyses) and providing clarifications (_e.g._, the meaning of vector $\boldsymbol{v}$). As the discussion period is ending soon, we would greatly appreciate it if you could provide feedback on whether you are satisfied with our response. We will be happy to address any remaining concerns.
>
> Best,
> Paper4429 Authors

---

### Official Review · Reviewer_ouEb · 2021-11-07

**Correctness:** 4
**Technical Novelty And Significance:** 3
**Empirical Novelty And Significance:** 3
**Recommendation:** 6
**Confidence:** 4

**Main Review:**

Strengths:

1. A clever approach to mitigate generator-discriminator dynamics in end-to-end training of Electra-like models by using a computationally efficient mixture of generators.
2. Strong empirical results on many downstream NLU tasks outperforming strong baselines.
3. Fairly thorough ablations for different model components.

Weaknesses:

1. It is unclear to me why the discriminator cannot just optimize gammas to be high for lower layers, thereby making it easier to optimize its own objective. Wouldn’t it, therefore, be better to have the gammas affect the MLM loss as well to help the lower layers learn better.
2. Following up on the previous point 1) Fig 3 (a) is counter-intuitive as to why layer 8 has the highest coefficient followed by 4 and then 6. I would have expected the discriminator to learn weights in the order 4 > 6 > 8 since the only gradient for these parameters comes from the discriminator loss and not the generator MLM loss (b) how robust is the mixture weights to restarts with different seeds and what if we trained from scratch with fixed mixture weights using what was obtained at the end of 120k steps in Fig 3 (a)?
3. Although computationally expensive, it may be useful to establish a performance “upper bound” by training M different generators.

**Summary Of The Paper:**

The paper presents an extension to Electra-like pre-training strategies that use MLM generator heads positioned at different layers throughout the networks and a single discriminator. Mixture-weights for different MLM outputs are also learned end-to-end via the gumbel-softmax estimator. The authors present evidence that generator-discriminator dynamics are impacted significantly by the generator’s capacity and architecture. The proposed idea, therefore, is to have the discriminator operator on top of a “mixture” of discriminators of different capacities. The authors make this mixture efficient by sharing the generator backbone and having separate MLM heads at different layers of the network and only learning the mixture weights of the representations produced at each layer.

**Summary Of The Review:**

This is a meaningful contribution to Electra-like pretraining approaches and provides fairly significant downstream improvements.

---

> ### Author Response · Authors · 2021-11-20
> **Response to Reviewer ouEb**
>
> Thanks for your review!
>
> Regarding the weaknesses:
> 1. **Why the discriminator cannot optimize gammas to be high for lower layers?** We note that the mixture weights are trained with the _reversed_ gradient from the discriminator, so they will be updated to always _maximize_ the discriminator loss (_i.e._, it is a form of adversarial training). Without the gradient reversal operation (_i.e._, the original discriminator gradient is backpropagated to train the mixture weights), the mixture weights for lower layers would have been high to minimize the discriminator loss. The learning target of the mixture is to increase the discriminator loss to compose challenging signals for training the discriminator but not to impact the MLM loss on the generator side. The lower layers of the generator are trained by its own MLM loss, not by the reversed gradients from the discriminator.
> 2. **Why are the mixture weights in Figure 3 (a) in the order of 8 > 4 > 6?** Since the mixture weights are trained to _maximize_ the discriminator loss, the deepest layer (8th) will have the highest average weights in composing the most challenging signals for the discriminator. As for why the 4th layer (instead of the 6th layer) gets the second highest average weight, we hypothesize that this is because layer 6 and layer 8 capture some shared linguistic aspects (as can be seen in Table 4 where the replaced tokens by the 6-layer and 8-layer MLMs are somewhat similar). Given that the 8th layer already has a high mixture weight, the 6th layer will not play a significant role in the output.
> 3. **How robust is the mixture weights to restarts with different seeds?** Note the pretraining cost is prohibitive for us to conduct full studies of variance in the pretraining step for each model variant (_i.e._, pretraining them five to ten times each is too costly). However, we did experiment with a handful of different random seeds for our AMOS base model, and the resulting mixture weights are almost the same throughout pretraining.
> 4. **What if we train from scratch with the fixed mixture weights learned at the end?** We have conducted this experiment as a new ablation and reported its performance in Appendix F (Table 9, w. final mix weights). We note that the mixture weights are computed based on the encoder output embeddings as shown in Equation (3), and each token will have different mixture weights given different token embeddings (the curves in Figure 3 (a) show the average mixture weights over all tokens in the batches at different training steps). Therefore, to train the discriminator with fixed mixture weights, we have to use the frozen converged AMOS generator (otherwise, any updates in the encoder will change the mixture weights which are determined by the encoder outputs, so we cannot train the generator from scratch if we want to fix the mixture weights). Such a setup leads to significantly worse results, confirming the necessity of gradually adapting the curriculum based on the discriminator's state throughout pretraining.
> 5. **Performance with different generators.** We conducted the experiment using three separate generators (4-layer, 6-layer and 8-layer) and reported it as an ablation in the original paper (Table 3, w. separate MLM gen.). It slightly underperforms AMOS and we hypothesize that this is because allowing the multiple generators to share a backbone (as AMOS multi-layer MLM does) could be beneficial for their MLM learning, similar to the advantage of multi-task learning.

---

> ### Author Response · Authors · 2021-11-27
> **Follow Up**
>
> Dear Reviewer ouEb,
>
> Thanks again for your review. We have tried our best to address your concerns by updating our manuscript (_e.g._, incorporating the new ablation you asked for) and providing clarifications (_e.g._, how and why the mixture weights are trained in an adversarial manner). As the discussion period is ending soon, we would greatly appreciate it if you could provide feedback on whether you are satisfied with our response. We will be happy to address any remaining concerns.
>
> Best,
> Paper4429 Authors

---

### Official Review · Reviewer_Hrj2 · 2021-11-08

**Correctness:** 3
**Technical Novelty And Significance:** 1
**Empirical Novelty And Significance:** 2
**Recommendation:** 3
**Confidence:** 5

**Main Review:**

  Questions:

1- Figure 1: the generator with more layers (8) gains lower training loss during pretraining, meaning it can better recover the masked token, which means during sampling it should recover the original token. In other words, the replaced-token sequence should have less replaced tokens and it should be an easier task for the discriminator!

2- Are the MLM heads shared across different layers of generator?

3- Table 2: the ablation on stop-gradient is only evaluated on MNLI and Squad which should degrade in performance. Is this ablation studied on other GLUE tasks as well?

4- the author proposed using gumble-softmax to enable gradient backpropagation from discriminator. In ELECTRA paper, reinforcement learning is used to leverage this. Is the proposed approach can be used using RL too? what is the performance on downstream task?

5- The author mentioned that the discriminator loss is used to train the mixture weights of MLM output to combine a  more difficult signal for discriminator, but it will not update the MLM embeddings. This is confusing, because generator is trained jointly with discriminator, and MLM embeddings are trained via two gradients, one from discriminator, and the other from MLM pretraining of generator! Moreover, in Figure 2, it is shown that discriminator gradient are backpropagated through generator too. In table 2, it is shown that using adv. MLM hurts MNLI matched performance by 0.3 points, whereas other tasks are untouched. what is the performance of this setting on other GLUE tasks?

6- Table 1: why the number of parameters in Base++ setting is larger than Base one?

7- Table 2: all the ablation results are evaluated on MNLI and squad tasks. however, I think all GLUE tasks should be considered in this study to understand the actual contribution of different components of the proposed AMOS.

8- Table 2: in layer switch and random layer configuration, are all MLM layers are pretrained equally?

9- Table 2: w. separate MLM gen. configuration shows only 0.1 performance decrease on MNLI-matched compared to -stop grad setup. what is the performance on this setting on other GLUE tasks?

10- Figure 3(b): the discriminator accuracy with 4-layer generator have a smooth increase with higher performance than AMOS, which does not use any adversarial training, which seems an intuitive learning curriculum too. what is the explanation on this?

**Summary Of The Paper:**

This paper present a method for pretraining language model using generator-discriminator training. Similar to ELECTRA model which used a MLM model as generator, which corrupts input text and replace some tokens with Masked token, and to train a different language model to detects the replaced tokens. Different from ELECTRA, they proposed to use multiple MLM generator models, which replace tokens at different difficulty levels. This way, the discriminator is forced to learn higher level of language. In order to allow gradient back-propagation from discriminator to multiple generator, they employed gumble-softmax. The evaluation results on GLUE and Squad indicates better performance than previous arts.

**Summary Of The Review:**

The proposed adversarial approach combined with the multi-generator signal which is composed from different layers of a generator provides a curriculum learning approach to train a better discriminator for downstream tasks. Despite better results on GLUE and Squad tasks compared to previous arts, the extent to which the different component of AMOS contribute to the results are less studied in the ablation. Only two tasks (MNLI and Squad) are selected for ablation of adversarial learning, mixture weights, gradient stopping, etc. The evaluation results can be extended to SuperGLUE as well.

---

> ### Author Response · Authors · 2021-11-20
> **Response to Reviewer Hrj2**
>
> Thanks for your review. As listed in the general response, we have updated the paper with more experimental results, including discussions of ablations results on all GLUE tasks.
>
> To your questions:
> 1. **Using deeper generator makes the discriminator task easier?** This is not always the case. The difficulty of discriminator learning is determined by not only the amount of replaced tokens, but also their plausibility. As in Table 4, the replaced tokens by deeper generators are more plausible than by shallower generators and are more difficult to recognize, as reflected by the higher discriminator loss on the replaced tokens shown in Figure 1 (a). Also, an imbalanced classification problem (_e.g._, with way more positives than negatives) as in our case the signals provided by the 8-layer generator that contain very few replaced tokens, generally makes it harder to detect the minority class.
> 2. **Are the MLM heads shared across different layers of generator?** Yes, the multiple MLM heads share the same set of parameters. We have explicitly mentioned this in the revision.
> 3. **Ablations on other GLUE tasks.** We conducted ablation studies on MNLI and SQuAD because they are the most stable
> and representative downstream evaluations. Many previous papers also performed ablation studies on one or two GLUE tasks (MNLI was always included) instead of all GLUE tasks (_e.g._, Table 8 in BERT [1]; Tables 1, 2 & 4 in RoBERTa [2]; Table 4 in DeBERTa [3]). The small tasks in GLUE, (_e.g._, CoLA, RTE, MRPC) are very unstable as observed by various pretraining research. The ablation results on them are less informative. That being said, we extended all ablation studies to all GLUE tasks in Appendix F, but would like to note that the smaller task results are mainly for reference.
> 4. **Using RL instead of Gumbel-Softmax.** Please refer to our response to Reviewer UXyk and Appendix H.
> 5. **Are MLM embeddings trained via two gradients?** We like to clarify that MLM embeddings are trained with only one gradient (backpropagated from the MLM objective) to ensure that the MLM learning is not disturbed by the discriminator signal, as discussed in the last paragraph of Section 3.3. The gradient backpropagated from the discriminator side is only used to train the mixture weights and the stop gradient operator isolates it from training MLM embeddings. Using two gradients to train MLM embeddings is studied in Table 2 as an ablation (+ adv. MLM). We found that the +adv. MLM run has to use a quite small multiplier on the gradients (0.1 in our experiments) from the discriminator to the MLM embeddings, otherwise the pretraining becomes unstable (_e.g._, using 0.2 multiplier weight resulted in a divergence). Due to the small gradient multiplier, this ablation has close performance with AMOS, but it is much more unstable.
> 6. **Why does the Base++ model have more parameters than the Base model?** As mentioned in the
> first paragraph of Section 4, our _Base_ model uses 32,768 uncased BPE vocabulary while _Base++_ model uses 64,000 cased BPE vocabulary, which causes the difference in their model parameters. These vocabulary sizes follow previous studies (_e.g._, Section 4.4 in RoBERTa [2] and Table 13 in DeBERTa [3]). The added token embedding parameters barely increase computing cost as the Transformer network size does not change.
> 7. **Ablations on all GLUE tasks.** Please see our discussions in Appendix F.
> 8. **Are all MLM layers are pretrained equally for ablations?** The multiple MLMs used in ablations are trained under the exact same setup with the AMOS model---jointly trained throughout pretraining with shared MLM head parameters.
> 9. **Ablations on other GLUE tasks.** Please see our answers to question 3.
> 10. **Curriculum learning of using a 4-layer generator.** Using a 4-layer generator (without adversarial mixture) also provides a curriculum as the generator gradually learns better and constructs more difficult signals. As in Section 3.1, this is one of the key advantages of ELECTRA-style pretraining. Then in Section 3.2, we show that a curriculum not adjusting to the discriminator training may not provide the most informative training signals. In Figure 3 (b), the discriminator can easily overcome the increased difficulty from a 4-layer generator: the signals may not be challenging enough later in training. AMOS progressively adapts the mixture weights over multiple MLMs to provide more challenging signals according to the latest discriminator state, leading to a lower discriminator accuracy than using a 4-layer generator in Figure 3 (b). With the new adversarial learning curriculum, AMOS enjoys better pretraining efficiency and achieves better downstream performance.
>
> References:
> [1] Devlin et al. BERT: Pre-training of Deep Bidirectional Transformers for Language Understanding.
> [2] Liu et al. RoBERTa: A Robustly Optimized BERT Pretraining Approach.
> [3] He et al. DeBERTa: Decoding-enhanced BERT with Disentangled Attention.

---

> > ### Comment · Reviewer_Hrj2 · 2021-11-22
> > **comment on author response [1/N]**
> >
> > Thanks for providing responses to the questions.
> > Regarding comparison on all GLUE tasks in Table 9, It would be more clear and understandable to add average performance column on GLUE as well. In table 1, it is shown that AMOS outperforms by 1 points compared to prior arts. However, in Table 7, Multi-MLM and Adversarial setup have very close performance or better one on different GLUE tasks. In other words, what is the overall gain of AMOS in terms of average performance on GLUE compared to Multi-MLM and Adversarial setup?

---

> > > ### Author Response · Authors · 2021-11-25
> > > **Response to Reviewer Hrj2 Comments**
> > >
> > > Thanks for your comments. We would like to note again that the GLUE average score is _not_ a reliable signal for ablation study because the small GLUE tasks (_e.g._, CoLA, RTE, MRPC) are very unstable and the model performance on those tasks is heavily influenced by randomness in both pretraining and fine-tuning stages. For example, Sellam et al. [1] reported that two checkpoints pretrained by exactly the same BERT algorithm with only random seeds being different can have a maximum of 4 points difference on CoLA, 5 points difference on MRPC, and 10 points difference on RTE, even after taking the median/average of multiple fine-tuning runs. We have similar observations in our experiments. Such high randomness of small GLUE tasks makes them (and thus the GLUE average score) unreliable for comparing the effectiveness of different design choices. On the contrary, large tasks like MNLI are much more stable and reliable to reflect the real model effectiveness (_e.g._, two checkpoints pretrained by the same algorithm with different random seeds have 0.6 points difference at most on MNLI according to Sellam et al. [1]), so we mainly relied on large tasks to make decisions among different design choices.
> > >
> > > To make the small task fine-tuning more stable (and thus the GLUE average score a bit more meaningful for ablation study), in addition to the results in the paper, we would like to present a new set of ablation results where the small task fine-tuning (CoLA, RTE, MRPC, STS-B) starts from the same model's MNLI fine-tuned checkpoint (instead of the pretrained checkpoint). This is a standard trick to help stabilize and improve the performance on small GLUE tasks (_e.g._, RoBERTa paper [2] Section 5.1 and ELECTRA paper [3] Appendix B reported that they achieved more stable and better performance on RTE, STS-B and MRPC tasks by adopting this trick). It is also widely used in many GLUE leaderboard submissions. In the following table, the large task (MNLI, QQP, QNLI and SST-2) results are identical to those in Table 9. The small task (CoLA, RTE, MRPC, STS-B) results are generally improved over those in Table 9 using this continuing-from-MNLI trick.
> > >
> > >
> > > | Method | MNLI-m/mm | QQP | QNLI | SST-2 | CoLA | RTE | MRPC | STS-B | AVG |
> > > | ------ | ------ | ------ | ------ | ------ | ------ | ------ | ------ | ------ | ------ |
> > > | AMOS | **88.9**/**88.7** | **92.3** | 93.6 | **94.2** | 71.1 | 89.2 | 91.9 | 91.8 | **89.1** |
> > > | w. random layer |  88.6/88.3 | 92.2 |	93.2 |	93.9 |	68.6 |	87.4 |	91.4 |	**92.0** |	88.4 |
> > > | w. layer switch | 88.6/88.4 | 92.2 |	93.0 |	94.0 |	67.0 |	87.7 |	**92.2** |	91.8 |	88.3 |
> > > | - adv. train | 88.7/88.5 | **92.3** |	93.2 |	**94.2** |	**71.7** |	**89.5** |	91.4 |	91.8 |	**89.1** |
> > > | + adv. MLM | 88.6/**88.7** |	92.2 |	93.5 |	93.8 |	67.1 |	88.1 |	91.4 |	91.7 |	88.3 |
> > > | w. final mix weights | 88.4/88.0 | 92.2 |	93.0 |	93.6 |	71.2 |	85.6 |	91.4 |	91.8 |	88.4  |
> > > | - stop grad. | 88.5/88.6 | **92.3** |	**93.9** |	93.9 |	69.6 |	86.6 |	91.4 |	91.6 |	88.5 |
> > > | w. separate MLM gen. | 88.8/**88.7** | **92.3** |	93.2 |	94.0 |	70.6 |	88.1 |	90.7 |	91.6 |	88.7 |
> > > | w. three 4-layer gen. | 88.6/88.4 | 91.9 | 92.8 |	93.9 |	68.2 |	86.6 |	91.7 |	91.4 |	88.1 |
> > > | 4-layer gen. | 88.4/88.2 | 92.2 |	93.1 |	**94.2** |	68.0 |	87.4 |	91.7 |	91.6 |	88.3 |
> > > | 6-layer gen. | 88.6/88.3 | 92.2 |	93.2 |	93.2 |	69.8 |	87.0 |	91.2 |	91.7 |	88.3 |
> > > | 8-layer gen. | 88.3/88.1 | 92.1 |	93.0 |	93.5 |	67.6 |	86.6 |	91.7 |	91.7 |	88.1 |
> > > | 12-layer gen. | 87.8/87.5 | 92.1 | 92.7 |	92.7 |	70.4 |	84.5 |	90.9 |	91.6 |	87.8 |
> > >
> > > AMOS improves over most ablated versions by 0.5+ GLUE average score, demonstrating the importance of each model component in our framework. Although the "-adv. train" ablation gets the same GLUE average score with AMOS, AMOS has better or same performance than "-adv. train" on all large tasks (MNLI, QQP, QNLI, SST-2 and SQuAD 2.0) which confirms the benefits of adversarial training; the advantage of "-adv. train" over the 4/6/8-layer gen. models reflects the benefits of training with a mixture of generator signals. The two components together (adversarial training + mix of signals) are the sources of AMOS's effectiveness. Despite better overall GLUE score obtained in this table, our preference is to maintain the single-task fine-tuning setting in our paper, to avoid potential confusions (_e.g._, readers may compare our GLUE scores in the above table to previous research's single-task fine-tuning results and perceive unfair empirical advantages of AMOS over previous work).
> > >
> > > References:
> > > [1] Sellam et al., The MultiBERTs: BERT Reproductions for Robustness Analysis.
> > > [2] Liu et al. RoBERTa: A Robustly Optimized BERT Pretraining Approach.
> > > [3] Clark et al. ELECTRA: Pre-training Text Encoders as Discriminators Rather Than Generators.

---

> > > ### Author Response · Authors · 2021-11-27
> > > **Follow Up**
> > >
> > > Dear Reviewer Hrj2,
> > >
> > > Thanks again for your review and your comments. We have tried our best to answer all your questions by updating our manuscript and providing new results (_e.g._, GLUE average scores of all ablations). As the discussion period is ending soon, we would greatly appreciate it if you could provide feedback on whether you are satisfied with our response. We will be happy to address any remaining concerns.
> > >
> > > Best,
> > > Paper4429 Authors

---

### Official Review · Reviewer_UXyk · 2021-11-08

**Correctness:** 3
**Technical Novelty And Significance:** 4
**Empirical Novelty And Significance:** 4
**Recommendation:** 8
**Confidence:** 4

**Main Review:**

There are several strengths from the paper that make me believe it is a good paper to be published in ICLR 2022:
1. The paper made a significant contribution to idea of using adversarial training as part of the self-supervision signal for language learning.
2. The paper made a impactful finding for practicing adversarial training, that mixture of signals at different depth of of the generator can stabilize ELECTRA-style models trained adversarially using Gumble-Softmax relaxation.
3. The experiments in the paper demonstrated the superiority of adding adversarial training to a self-supervision framework, in that significant improvements can be obtained for similar-sized networks.
4. Good set of ablation studies to show that each component of the model is necessary, especially because the entire model already has many moving parts in addition to adversarial training.

With the strengths being said, I hope to also point out that the paper's application of adversarial training is one attempt in many possibilities, and in many cases it is not clear where the improvements come from. For example:
1. The paper pointed out that ELECTRA framework [1] explored the idea of using REINFORCE [2] as the the way of adding adversarial training signals to the model but observed degenerated results. Compared to this, the paper made 2 changes to the model: 1) using Gumble-Softmax instead of REINFORCE, and 2) using mixture-of-signals instead of straightforward gradient back-propagation. It is hard to know here which one of these actually made the adversarial setup useful. Is it possible to run an ablation study using the combination of REINFORCE and mixture-of-signals to verify whether Gumble-Softmax relaxation is the reason for it to work?
2. There are many past papers that apply the discriminator to some internal representations of the generator instead of on the softmax outputs of the discrete text signal (for example, [3][4][5]). In the practices of GAN for text, these are proven to be more stable than both Gumble-Softmax relaxation and REINFORCE. What are the reasons for the paper to choose Gumble-Softmax relaxation instead? Please discuss.

References:

[1] Kevin Clark, Minh-Thang Luong, Quoc V. Le, Christopher D. Manning, ELECTRA: Pre-training Text Encoders as Discriminators Rather Than Generators, ICLR 2020

[2] Lantao Yu, Weinan Zhang, Jun Wang, Yong Yu, SeqGAN: Sequence Generative Adversarial Nets with Policy Gradient, AAAI 2017

[3] Yizhe Zhang, Zhe Gan, Kai Fan, Zhi Chen, Ricardo Henao, Dinghan Shen, Lawrence Carin, Adversarial Feature Matching for Text Generation, ICML 2017

[4] Jake Zhao (Junbo), Yoon Kim, Kelly Zhang, Alexander M. Rush, Yann LeCun, Adversarially Regularized Autoencoders, ICML 2018

[5] Sandeep Subramanian, Sai Rajeswar Mudumba, Alessandro Sordoni, Adam Trischler, Aaron C. Courville, Chris Pal, Towards Text Generation with Adversarially Learned Neural Outlines, NeurIPS 2018

**Summary Of The Paper:**

The paper proposes a method to mitigate the difficulty of training deep adversarial auto-regressive generators in the ELECTRA self-supervised framework, by extracting the hidden representation from each layer of the deep generator network and feeding it through a mixture of representations function. The resulting model, named adversarial mixture of signals (AMOS), combined with the use of Gumble-Softmax relaxation, effectively stabilized GAN-style adversarial training for ELECTRA-style frameworks. This enabled the possibility of learning better generators and discriminators, by making it possible to feed gradients from the discriminator directly to the generator. The paper provided an extensive set of experiments for self-supervised language learning using GLUE and SQuAD tasks, for which AMOS showed state-of-the-art results compared to similarly-sized alternatives. Furthermore, extensive ablation studies are provided to show that both curriculum setup (as in the ELECTRA-style frameworks) and adversarial setup (as in AMOS, which is new in the paper) improve the results. These studies also verify the effectiveness of stabilizing training using the mixture-of-hidden-representations setup, especially for deeper generators.

**Summary Of The Review:**

Novel idea of stabilizing ELECTRA-style language learning using a mixture of signals at different depth of generator hidden representations. Significant contribution to using adversarial training as part of the signal for self-supervised language learning. Good experimental results of the proposed model. Good set of ablation studies for each part of the model. Some unclear reasoning in the choice of adversarial training techniques compared to previous papers.

---

> ### Author Response · Authors · 2021-11-20
> **Response to Reviewer UXyk**
>
> Thanks for your review! We have added more discussions and additional experimental results in the updated version, as listed in the general response.
>
> Regarding your questions:
> 1. **Using Gumbel-Softmax vs. using REINFORCE**. We chose to use Gumbel-Softmax in our model because (1) it is simpler than the reinforcement learning setup as it is almost a drop-in replacement of the original softmax computation with only one additional hyperparameter (the temperature $\tau$), (2) it is more efficient which is critical for pretraining, and (3) reinforcement learning in the ELECTRA-style pretraining setting introduces an enormous action space, as discussed in ELECTRA paper's Appendix F, which we believe may require more sophisticated techniques and careful designs to obtain improvements. Our exploration is the first step to study constructing pretraining signals based on the model's state throughout pretraining via an adversarial learning curriculum, and our results showed positive evidence that this is a promising new research direction.
> 2. **Using REINFORCE with mixture-of-signals**: We doubt we can do a better job in using REINFORCE to connect the generator training with the discriminator signal than Clark et el. did in ELECTRA (not to mention our mixture-of-signal generator is more sophisticated than ELECTRA's single generator). Again, our choice of Gumbel-Softmax is not because it is necessarily more effective than any potential reinforcement learning technique (we believe this needs more future studies) but mainly because of its simplicity and efficiency. We indeed have conducted ablation studies to analyze the respective benefits of using mixture-of-signals and using Gumbel-Softmax for adversarial learning: In Table 2, the gap between "-adv. train" (from the Adversarial Setup group) and "4/6/8 layer gen." (from the Backbone group) reflects the contribution of mixture of signals; the gap between AMOS and "-adv. train" reveals the gain of adversarial mixture learning using Gumbel-Softmax. In our added experiments (Table 9), the gap between "-adv. train" and "w. three 4-layer gen." reflects the benefit from mixing a more diverse set of training signals.
> 3. **Comparison with Gumbel-Softmax alternatives**. Thanks for pointing out the references; we have included them in the paper revision (please see Appendix H). We would like to note that as the first paper that studies adversarial curriculums for language model pretraining, we aim to propose a simple and effective framework using standard techniques to demonstrate that such a new direction is promising. While there may exist more sophisticated and advanced realizations for specific model components in our framework (_e.g._, using alternative approaches to Gumbel-Softmax for obtaining the gradient from the discriminator), how to best leverage these methods in the pretraining setting definitely will be an interesting future research topic.

---

> > ### Comment · Reviewer_UXyk · 2021-11-24
> > **Maintaining acceptance recommendation**
> >
> > I agree with most of the judgements made in the reponse, such as REINFORCE being unlikely to work better than Gumbel-Softmax relaxation.
> >
> > I also agree with the authors that feeding internal representations to the discriminator can be better done in some follow-up work given the rich results already contained in the current paper.
> >
> > It is also nice for the authors to offer updated manuscript to include references and discussions to prior works for GAN for text.
> >
> > I feel that the response strengthened the paper, and I would have increased the score to 9 if such an option is provided to the reviewers. I therefore recommend acceptance to the stronger side of score 8.

---

> > > ### Author Response · Authors · 2021-11-25
> > > **Response to Reviewer UXyk Comments**
> > >
> > > Thanks for your feedback! It is great that we share similar visions on potential future directions, and we hope that we could keep improving the understanding of language model pretraining in the future.

---

### Author Response · Authors · 2021-11-20
**General Response**

We sincerely thank the six reviewers for thoughtful comments! We have updated the paper according to reviewers' suggestions and we summarize the changes as follows:
1. Add GLUE test set results per Reviewer CBi9 (Appendix D).
2. Add pretraining efficiency and run time analyses per Reviewer CBi9 (Appendix E).
3. Extend the ablation studies to all GLUE tasks per Reviewer Hrj2 and show their standard deviations per Reviewer oMcQ. Include two new ablation models per Reviewers ouEb and CBi9 (Appendix F).
4. Add performance study with different numbers of MLM heads per Reviewers sJ99 and oMcQ (Appendix G).
5. Add a discussion section regarding the choice of Gumbel-Softmax and other related techniques per Reviewer UXyk (Appendix H).
6. Change the probing experiment setting per Reviewer oMcQ (Table 3 and Appendix I).
7. Add a societal impact section and details of computation environments per Reviewer oMcQ (Reproducibility Statement and Appendix J).

We plan to integrate some of these appendix contents into the main paper in our next version when space permits.

---

### Public Comment · ~Cheng-Han_Chiang1 · 2022-02-23
**Questions on Table 4**

Thanks to the authors for such an interesting paper!
I would like to ask what the generators used for the results in Table 4 are.
Are those separate generators trained jointly (like the procedure in Section 5.1 **Effect of Multi-Layer Generator Setup.**), or are they the same generator in AMOS but sampled from different layers?

---

> ### Public Comment · ~Yu_Meng1 · 2022-02-24
> **Response to Question**
>
> Hi,
>
> Thanks for your interest in our work! The experiments in Table 4 were conducted using the same multi-layer MLM as in AMOS and sampling replaced tokens from different layers of the multi-layer MLM. We also tried using separate generators of different depths and observed similar patterns (i.e., the replaced tokens from deeper MLMs being more plausible and harder to be detected). Please let us know if you have any further questions!
>
> Best,
> Paper4429 Authors

---

### Decision · Program_Chairs · 2022-01-20

**Decision:**

Accept (Poster)

**Comment:**

This paper received six reviews, consisting of three 8s two 6s and one 3.
The reviewers generally felt that the proposed Electra-like pretraining provided fairly significant downstream improvements.
Additional ablations were provided to during the author response period and other author responses were sufficient to cause scores to rise during the discussion period.
The vast majority of reviewers recommended accepting this paper and the AC also recommends acceptance.